# Dynamic Multimodal Evaluation with Flexible Complexity by Vision-Language Bootstrapping

**Yue Yang**[1,2,*], **Shuibai Zhang**[2,*], **Kaipeng Zhang**[2,†], **Yi Bin**[3],
**Yu Wang**[1,2], **Ping Luo**[2,4,†], **Wenqi Shao**[2,†]
[1]School of Artificial Intelligence, Shanghai Jiao Tong University   [2]Shanghai AI Laboratory
[3]Tongji University   [4]The University of Hong Kong

## ABSTRACT

Large Vision-Language Models (LVLMs) have demonstrated remarkable capabilities across multimodal tasks such as visual perception and reasoning, leading to good performance on various multimodal evaluation benchmarks. However, these benchmarks keep a static nature and overlap with the pre-training data, resulting in fixed complexity constraints and data contamination issues. This raises the concern regarding the validity of the evaluation. To address these two challenges, we introduce a dynamic multimodal evaluation protocol called Vision-Language Bootstrapping (VLB). VLB provides a robust and comprehensive assessment for LVLMs with reduced data contamination and flexible complexity. To this end, VLB dynamically generates new visual question-answering samples through a multimodal bootstrapping module that modifies both images and language, while ensuring that newly generated samples remain consistent with the original ones by a judge module. By composing various bootstrapping strategies, VLB offers dynamic variants of existing benchmarks with diverse complexities, enabling the evaluation to co-evolve with the ever-evolving capabilities of LVLMs. Extensive experimental results across multiple benchmarks, including SEEDBench, MMBench, and MME, show that VLB significantly reduces data contamination and exposes performance limitations of LVLMs.

## 1 INTRODUCTION

Large Vison-Language Models (LVLMs) (Achiam et al., 2023; Lu et al., 2024b; Research, 2023; Chen et al., 2024b; Liu et al., 2024a) have achieved unprecedented performance across a wide range of multimodal tasks such as visual perception (Goyal et al., 2017) and commonsense reasoning (Zellers et al., 2019). The impressive progress necessitates the creation of nuanced evaluations to track LVLM development and understand the capability boundary of LVLMs. It leads to the advancement of a series of evaluation benchmarks such as SEEDBench (Li et al., 2023), MMBench (Liu et al., 2023), and MME (Fu et al., 2023). The evaluation results are crucial in selecting suitable LVLM for various applications.

Despite the proliferation of LVLM evaluations, there are increasing concerns about the genuine capabilities of LVLMs (Laskar et al., 2024), largely due to two key challenges associated with current evaluation benchmarks. *1) Data contamination*. LVLMs are pre-trained on large datasets, often sourced from the internet. Unfortunately, many evaluation benchmarks are constructed from similar sources, leading to a high likelihood of overlap with training data, thus causing data contamination (Touvron et al., 2023; Chen et al., 2024a), as illustrated in Figure 1(a) and detailed in Section 3. It raises a critical concern: "Does the model genuinely perceive and understand the input, or is it merely memorizing it?" *2) Static dataset with fixed complexity*. As shown in Figure 1(b), existing benchmarks for LVLMs are manually collected (Xu et al., 2023; Li et al., 2023). Once constructed, they are static with a fixed complexity, making them inadequate to keep pace with the rapid de-

---

∗ Equal contribution,† Corresponding Authors

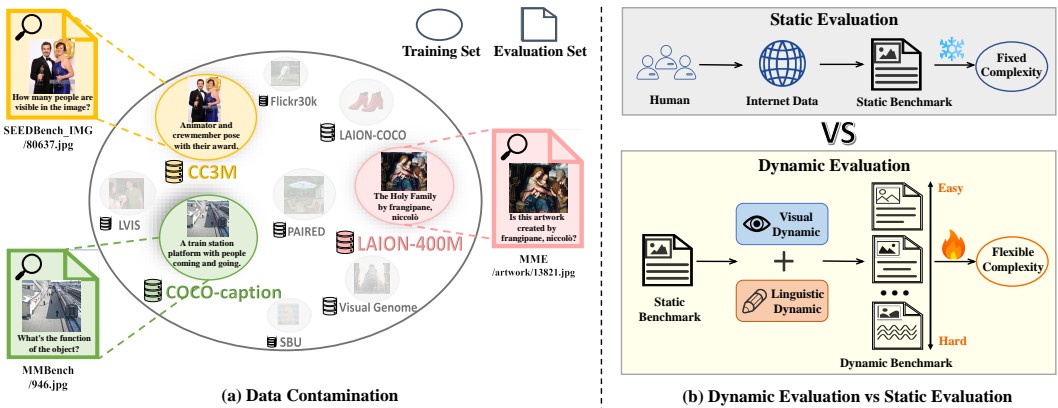

Figure 1: **(a)** shows that some images in evaluation sets can be exactly found in the training set and their corresponding questions can be solved by the captions of similar training images. **(b)** compares our dynamic multimodal evaluation with the previous static evaluation. We can see that dynamic evaluation can create various variants upon static benchmarks with flexible complexity.

velopment of LVLMs. To accurately assess LVLM performance boundaries, a dynamic, automated evaluation protocol with adjustable complexity is urgently needed.

However, it is challenging to develop a dynamic evaluation framework for LVLMs. Existing dynamic evaluation protocols apply to only Large Language Models (LLMs). For example, Dy-Val (Zhu et al., 2023) dynamically synthesizes test samples for LLMs based on directed acyclic graphs to combat data contamination. NPHardEval (Fan et al., 2023) generates new test samples for NP-hard math problems. Yet, the reliance on graph structure and math knowledge limits the applicability of these methods. Recently, MPA (Zhu et al., 2024a) dynamically creates new questions from the old ones for evaluating LLMs. However, building visual dynamics based on existing benchmarks requires manipulating the visual contents in the image while maintaining its essence, which poses a great challenge and remains unexplored.

In this work, we develop a dynamic multimodal evaluation (DME) protocol with flexible complexity by proposing Vision-Language Bootstrapping (VLB). VLB consists of a multimodal bootstrapping module and a judge module. The multimodal bootstrapping dynamically creates new visual question-answering (VQA) samples through various image and language bootstrapping strategies. These carefully designed strategies do not rely on predefined rules, making them applicable to various multimodal tasks. Meanwhile, the judge ensures that the newly generated samples maintain consistency with the original ones, *e.g.* preserving the correctness of the answer after bootstrapping.

To dynamically create new evaluation suites with flexible complexity, we propose various bootstrapping strategies with complexity control for both image and question. These strategies stem from practical situations in real user interactions, primarily simulating user diversity in two aspects: different visual attention and linguistic understanding. By bootstrapping a VQA sample with various transformations flexibly, we can build a series of dynamic variants with diverse complexities upon an existing benchmark. Experimental results show that the data contamination issues can also be remarkably reduced on these dynamic variants. Therefore, the proposed VLB can provide a comprehensive and robust assessment of LVLM capabilities, ensuring that the evaluation co-evolves with the ever-evolving abilities of LVLMs.

Our VLB is general enough to be applied to various existing benchmarks. In experiments, we employ VLB to bootstrap several representative evaluation benchmarks such as SEEDBench (Li et al., 2023) and MMbench (Liu et al., 2023). We evaluate various LVLMs including closed-source APIs (GPT-4o (OpenAI, 2024), Claude3-Sonet (Anthropic, 2024)), and 9 popular open-source LVLMs such as DeepSeek-VL (Lu et al., 2024a) and Yi-VL (Young et al., 2024). The takeaways of our key findings are as follows:

- All LVLMs exhibited performance decreases on our multimodal dynamic benchmarks, indicating existing limitations in current LVLMs towards adapting to different users with various identities. (Section 5.2 and 5.3)

- Through image bootstrapping strategies, we observed that distracting or focusing visual attention on the image will affect LVLMs' capabilities to understand the key image content related to the corresponding question. (Section 5.2)

- By employing language bootstrapping strategies on existing benchmarks, we found that current LVLMs exhibit increasing sensitivity as modifications escalate from words to sentences and finally to context changes. (Section 5.2)

- VLB can provide multiple dynamic benchmarks with flexible complexity via multimodal and multi-strategy composition. By combining more difficult strategies, we can increase the complexity of existing benchmarks, posing significant challenges for LVLMs, including advanced models like InternVL-2 and GPT-4o. (Section 5.3)

- LVLMs exhibit varying performance changes on our dynamic strategies across various tasks, especially sensitive to 'Instance Interaction', 'Text Understanding', and 'Spatial Relation' tasks. (Section 5.4)

Overall, the **contributions** of this paper are summarized as follows. 1) We delve into the data contamination issue of existing multimodal evaluation benchmarks and find a pronounced overlap between evaluation samples and pre-training data, which hinders the assessment of previous static benchmarks. 2) We propose a dynamic multimodal evaluation protocol called vision-language bootstrapping (VLB). VLB can evolve existing benchmarks with visual and linguistic dynamics, obtaining various variants with flexible complexity and reduced data contamination. 3) We perform comprehensive evaluations on a variety of popular LVLMs, indicating that the existing LVLMs still struggle to adapt to different user interactions and intent. Our code will be available at `https://github.com/yangyue5114/DME`.

## 2 RELATED WORK

**Data contamination.** Data contamination raises widespread concerns across evaluation benchmark datasets in both LLMs and LVLMs. Study (Dodge et al., 2021) reveals exact match contamination rates ranging from 2% to 50% across various benchmarks when compared to C4 pretraining data. Llama-2 (Touvron et al., 2023) reported over 16% of MMLU examples are contaminated, where 11% are seriously contaminated. Subsequently, the concern of data contamination extends to LVLMs. Recent work (Chen et al., 2024a) reveals the data leakage issue in LVLMs, they find some samples are leaked into LLMs' training corpora can be "recalled" with the textual questions and answers directly. For example, Sphinx-X-MoE (Gao et al., 2024) gets 43.6% on MMMU (Yue et al., 2024) without accessing images, surpassing its LLM backbone with 17.9%. But it is limited to the textual modality of LVLMs, leaving image modality unexplored and the specific contamination rate undetected. In section 3, we first verify a high rate of contamination in both visual and linguistic modalities, between static multimodal benchmarks and training sets. This severe contamination necessitates the development of dynamic evaluation to advance LVLMs more accurately.

**LVLM Evaluation.** With the rapid development of LVLMs, various benchmarks are proposed for evaluating LVLMs. Early single-task benchmarks, like ok-VQA (Marino et al., 2019), and MS-COCO (Chen et al., 2015), are proposed to evaluate coarse-grained ability. Later, current LVLM evaluation benchmarks aimed to provide relatively holistic evaluations for the overall reasoning capabilities of LVLMs, such as LVLM-eHub (Xu et al., 2023), MME (Yin et al., 2023), SEED-Bench (Li et al., 2023), MM-Vet (Yu et al., 2023), MMBench (Liu et al., 2023), and Llavebench (Liu et al., 2024b). However, the static nature of current benchmarks makes them susceptible to contamination and restricts them to a fixed level of complexity. Thus, there is an urgent need for dynamic evaluation methods to enable comprehensive assessments of LVLMs.

**Dynamic Evaluation.** Recently, some researchers have pioneered the exploration of dynamic evaluation. Study (Zhu et al., 2023) proposed DyVal to dynamically generate test samples based on the graph structure to combat data contamination. Similarly, NPHardEval (Fan et al., 2023) generates new evaluation samples for NP-hard mathematical problems. To extend dynamic evaluation to more diverse NLP tasks, Zhu et al. (2024a) further developed MPA, which employs LLM-based agents

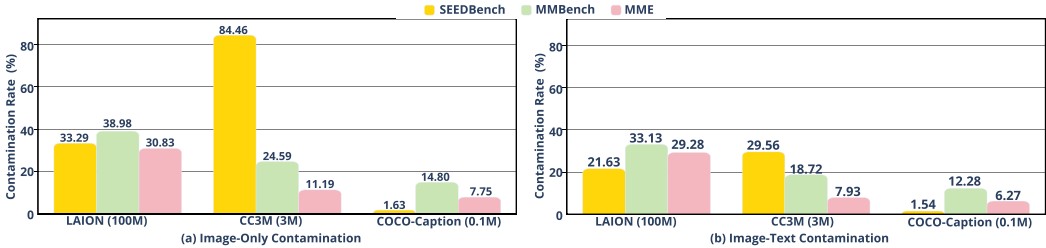

Figure 2: (a) Existing benchmarks have severe overlap on images with pre-training data. (b) Questions of the contaminated evaluation image can also be solved by the caption of similar images from the training set.

to transform existing problems into new ones. However, these dynamic evaluations only focus on LLMs and NLP benchmarks, leaving the dynamic evaluation of LVLMs unexplored. Previous multi-modal benchmarks (Shah et al., 2019; Gokhale et al., 2020; Selvaraju et al., 2020) have made attempts to rephrase VQA questions, but they can't be automatically applied to various formats of existing benchmarks. Our work is the first to design strategies for bootstrapping both visual images and language questions, to achieve dynamic evaluation on LVLMs. Based on an original benchmark, we can generate multiple variants with flexible complexity and lower contamination rates.

## 3   REVISITING DATA CONTAMINATION IN EXISTING BENCHMARKS

We explore two types of data contamination in multimodal evaluation benchmarks. **1) Image-only contamination.** We aim to detect how many images in the benchmark can be found in the pre-training data. To this end, we utilize the CLIPScore (Hessel et al., 2021) to measure the similarity between images from the evaluation and training set. In our pilot experiments, we find that if the CLIPScore between two images exceeds 0.9, it indicates high visual similarity. Thus, we adopt 0.9 as the threshold to determine visual contamination. The image-only contamination rate is calculated as the ratio of the number of contaminated images and the number of total images in the evaluation set. **2) Image-text contamination.** Beyond images, the question and answer of benchmark can also be contaminated. We extend ideas from NLP detection works (Li et al., 2024a) to identify this image-text contamination. For contaminated image pairs, we determine the question and answer contaminated if the answer can be directly inferred from the captions of the training image. In practice, we leverage GPT-4 to conduct this process.

As illustrated in Figure 2, we examine the two types of data contamination between three popular evaluation benchmarks: SEEDBench, MMBench, MME, and three widely used pre-training sets: LAION-100M (Schuhmann et al., 2021), CC3M (Changpinyo et al., 2021), COCO-Caption (Chen et al., 2015). The results reveal that each evaluation benchmark presents certain contamination rates across training datasets of various sizes, even with some reaching as high as 84.46% (image-only) and 33.13% (image-text). Note that the actual size of pre-training data far exceeds our detected maximum of 100M, which indicates that the actual contamination issue could be even more severe.

## 4   DYNAMIC MULTIMODAL EVALUATION

This section introduces our dynamic multimodal evaluation framework, Vision-Language Bootstrapping (VLB). Section 4.1 gives an overview. Section 4.2 and 4.3 present one component of VLB, multimodal dynamics: mage bootstrapping and language bootstrapping. The other component, the judge module, is in Section 4.4. The compositional bootstrapping is described in Section 4.5.

### 4.1   OVERVIEW

**Formulation of Dynamic Multimodal Evaluation (DME).** Let us begin with defining a dynamic multimodal evaluation for LVLMs. Given an original VQA sample $E_s = (I, Q, A)$ where $I, Q$

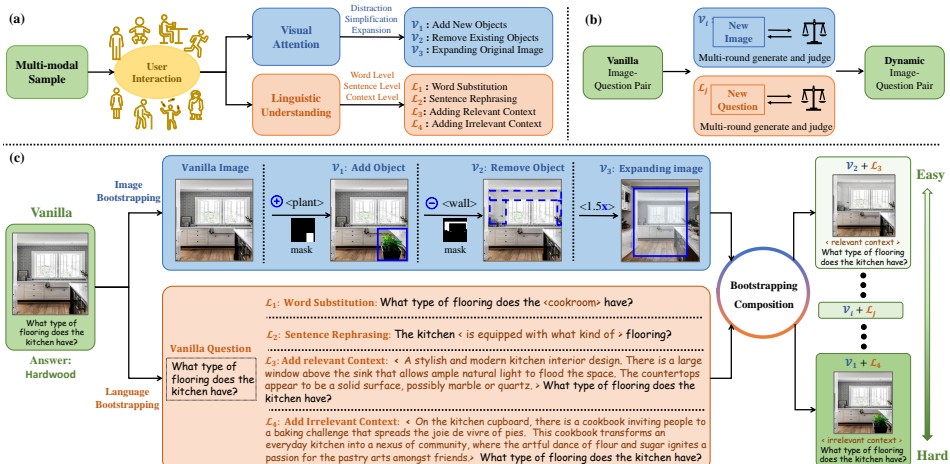

Figure 3: Illustration of our proposed dynamic multimodal evaluation framework, Vision-Language Bootstrapping (VLB). (a) demonstrates how we derive insights from real user interactions with LVLMs, where users possess different visual attention and language understanding from diverse identities. (b) highlights the role of VLB's judge module in ensuring that generated images and questions maintain consistent with the original. (c) provides an example of VLB transforming a sample through image and language bootstrapping. Additionally, VLB can generate new, increasingly complex samples through bootstrapping composition.

and $A$ denote the image, question, and ground truth answer, a dynamic sample can be represented as $E_d = (\mathcal{V}(I), \mathcal{L}(Q), A)$. Here we use the subscripts '$s$' and '$d$' to denote the original static and dynamic evaluation samples, respectively. $\mathcal{V}$ and $\mathcal{L}$ are transformations applied to the image and question. By comparing $E_s$ with $E_d$, we see that DME is implemented by image and language bootstrapping while preserving the consistency with the original sample.

**Challenges.** There are three major challenges in achieving dynamic multimodal evaluation. First, the DME framework should be generic enough to be used in various multimodal tasks instead of limited to a specific task like (Fan et al., 2023). Second, it is hard to alter images and questions while not affecting the correctness of the reference answer. In linguistics, GPT4 can be used as a powerful agent to rewrite questions while preserving the consistency with the original question by following instructions (Zhu et al., 2024a). However, there is no agent to achieve this in image transformation. Third, the DME framework should be flexible enough to control the complexity of newly generated evaluation samples, enabling the evaluation to co-evolve with LVLM development.

**Our Approach.** We propose a novel DME protocol, called vision-language bootstrapping (VLB). VLB consists of a multimodal bootstrapping module to generate new images and questions, and a judge module to maintain consistency with the original sample. As illustrated in Figure 3 (a), by simulating real LVLM's user interaction in visual attention and linguistic understanding, we design image (*i.e.*, $\mathcal{V} \in \{\mathcal{V}_1, \mathcal{V}_2, \cdots\}$) and language (*i.e.*, $\mathcal{L} \in \{\mathcal{L}_1, \mathcal{L}_2, \cdots\}$) bootstrapping strategies. Experiments show that the composition of $\mathcal{V}$ and $\mathcal{L}$ would yield dynamic evaluation samples with flexible complexity, an example is exhibited in Figure 3 (c).

## 4.2 Image Bootstrapping

Our VLB employs image bootstrapping to build visual dynamics. Since different users pose various identities and backgrounds, their levels of visual attention also vary. For instance, children's attention might be more easily distracted by irrelevant objects in images than educated adults. Thus, we simulate disturbing or focusing visual attention for image bootstrapping: adding new objects, removing existing objects, and expanding original images, as Figure 4. Each strategy is associated with a difficulty level (*i.e.* 'Easy' or 'Hard').

$\mathcal{V}_1$: **Adding New Objects (Hard).** Given the original sample $E_s = (I, Q, A)$, $\mathcal{V}_1$ aims to insert a new object into the image $I$ while maintaining that the answer $A$ is still correct for generated image

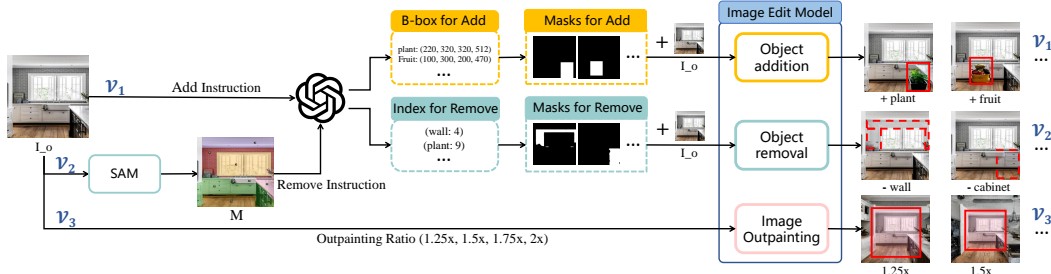

Figure 4: Image bootstrapping strategies: Starting from an original image, route $\mathcal{V}_1$, $\mathcal{V}_2$, $\mathcal{V}_3$ represents the process of adding new objects, removing existing objects, and expanding original images.

$\mathcal{V}_1(I)$. To this end, we integrate GPT-4V (Achiam et al., 2023) and the advanced image editing model PowerPaint (Zhuang et al., 2023) in two steps. First, GPT-4V is prompted with a carefully designed add-instruction (detailed in Appendix A.1) to return a set of insertable objects. Each object is in the form of (object name: bounding box). Second, we transformed the bounding box into mask, which together with the object name and original image $I$, are fed into the PowerPaint to generate the new image $\mathcal{V}_1(I)$. Adding objects into an image introduces irrelevant visual content, which would distract visual attention and may be harder than the vanilla sample $E_s$.

$\mathcal{V}_2$: **Removing Existing Objects (Easy).** Simplifying the visual content by removing objects poses a substantial challenge because GPT-4V is relatively weak in fine-grained visual grounding. To tackle this problem, we extend the idea of SoM (Yang et al., 2023) that unleashes the visual grounding capabilities of GPT-4V by providing semantic marks. Leveraging SAM (Kirillov et al., 2023), we obtain semantic masks $\mathbf{M}$ of all objects within an image. We then assign each object with a serial number. Afterward, we prompt GPT-4V with remove-instruction to give the removable objects, each formatted as (object name, serial number). Finally, the image $I$, object name, and mask corresponding to the serial number are fed into PowerPaint (Zhuang et al., 2023) to obtain the new image. Since removing objects from the image would omit superfluous visual content simplifying visual attention, $E_d$ would be easier than the vanilla $E_s$ under $\mathcal{V}_2$.

$\mathcal{V}_3$: **Expanding Original Images (Hard).** $\mathcal{V}_3$ employs image outpainting to extend the background of the image while not affecting the core elements within the image relevant to the question $Q$. Since expanding the image also introduces irrelevant visual content to the problem, $E_d$ would be more challenging than the vanilla $E_s$ under $\mathcal{V}_3$. We set the extension ratio $r = 1.5$ for the main experiments. Intuitively, varying the outpainting ratio $r$ mimics the effect of observing objects from different distances. We provide an ablation study on the effect of outpainting ratios in Section 5.4.

### 4.3 LANGUAGE BOOTSTRAPPING

Besides visual modality, language understanding is also crucial for LVLMs. We construct language bootstrapping by stimulating different linguistic expressions of users with various identities and backgrounds. Four strategies from different levels of linguistic expressions: word level, sentence level, context level, are employed. Figure 3 illustrates the transformed samples of four paradigms $\mathcal{L}_1, \mathcal{L}_2, \mathcal{L}_3, \mathcal{L}_4$ based on the original question $Q$. We also assigned a difficulty level with 'Easy' or 'Hard' for each language bootstrapping strategy.

$\mathcal{L}_1$: **Word Substitution (Hard).** Given that each user of LVLMs possesses unique vocabulary preferences and habits, $\mathcal{L}_1$ simulate these variations by replacing words with synonyms or contextually similar expressions while preserving the core concept for question $Q$ by following (Zhu et al., 2024b). LLM pre-training usually has limited coverage of phrasing styles in instructions. Therefore, $\mathcal{L}_1$ would increase the difficulty of the original problem.

$\mathcal{L}_2$: **Sentence Rephrasing (Hard).** Considering users with different identities might express the same question with various phrasing styles. For example, the question "How does photosynthesis work?" would become "How do plants make food from sunlight?" for a casual user, while turning into "What are the key differences between the light-dependent and light-independent reactions in

photosynthesis?" for a researcher. To this end, $\mathcal{L}_2$ employs role-playing (Wang et al., 2023) to rephrase the original question $Q$ into new ones.

$\mathcal{L}_3$: **Adding Relevant Context (Easy).** $\mathcal{L}_3$ aims to introduce visual-related text to the original question $Q$. The generated context should contain more hints to help understand the image. However, it should be ensured that the original question cannot be solved directly with the added context. The strategy reflects the situation when users understand the image partly. We employ GPT-4V to generate descriptions of the image via our carefully designed prompts (Appendix A.1). Then we add the generated description to $Q$, obtaining the new question $\mathcal{L}_3(Q)$. By design, the dynamic variant $E_d$ would be easier than the original $E_s$.

$\mathcal{L}_4$: **Adding Irrelevant Context (Hard).** Unlike $\mathcal{L}_3$, $\mathcal{L}_4$ will supplement the question $Q$ with visual-related text which cannot help answer the question. Here we also use GPT-4V to create context. Since the irrelevant context would distract LVLMs, the new sample $E_d$ would become harder than the original $E_s$.

## 4.4 Judge Module

DME should create new samples while maintaining essential concepts of the original one. Although our VLB employs strict criteria in the bootstrapping process, the transformation may still break the consistency. For instance, the added objects in $\mathcal{L}_1$ might shelter key visual content relevant to the problem. To tackle this issue, we incorporate a judge module to ensure consistency between the dynamic variant and the original sample.

Specifically in Figure 3(b), given the static sample $E_s = (I, Q, A)$ and the corresponding dynamic variant $E_d = (\mathcal{V}(I), \mathcal{L}(Q), A)$, the judge module is informed what modification is conducted by $\mathcal{V}(I)$ and $\mathcal{L}(Q)$, and is asked to check whether $A$ is still correct for the newly generated image and question. Here we designed different judge prompts to check each bootstrapping strategy, detailed prompts are in Appendix A.2. Similar to MPA (Zhu et al., 2024a), the judge operates in an adversarial manner and returns a 'Yes' or 'No' verdict. If the response is 'No', the sample will be regenerated until it passes the assessment. If the sample does not pass after five attempts, the original sample is used instead. In practice, we use InternVL-2, an open-source powerful LVLM (Chen et al., 2023; 2024b), as a capable and affordable judge. What's more, we further validate the effectiveness of our judge module by the human verification in Section A.5

## 4.5 Compositional Bootstrapping

Due to every single VLB strategy for image and question being atomic, we can investigate two kinds of bootstrapping composition with flexible complexities. 1) Paired multimodal composition. We can compose visual bootstrapping $\mathcal{V}$ and linguistic bootstrapping $\mathcal{L}$, into a paired multimodal dynamic sample $E_d = (\mathcal{V}_i(I), \mathcal{L}_j(Q), A)$, obtaining a total of 12 dynamic variants. 2) Multi-strategy composition. We can also stack multiple image bootstrapping strategies on a single image or multiple language bootstrapping strategies on the question, composing a multi-strategy dynamic sample like $E_d = (\mathcal{V}_3(\mathcal{V}_1(I)), \mathcal{L}_4(Q), A)$.

Since each single VLB strategy possesses different levels of complexity, the above two kinds of compositions can effectively construct different variants varying in complexity, to assess the robustness and adaptability of LVLMs and explore models' upper and lower limits in performance across different benchmarks. The details and results of the strategy compositions are in Section 5.3.

## 5 Experiment

### 5.1 Experiment Setup

**Tasks and Datasets.** We selected five popular benchmarks to assess current LVLMs, encompassing Yes/No Questions (MME), Multiple Choice Questions (MMBench, SEEDBench), and Visual Question Answering (MMvet, LLaVABench). These benchmarks include a broad spectrum of cognitive and comprehension tasks. In Section 5.2 and 5.3, we employ three comparable datasets in terms of size: MME, MMBench (30%), and SEEDBench (10%) as the experimental datasets. Then we extend our dynamic strategies to the full set of MMBench, MMvet, and LLaVABench in Section 5.3

Table 1: Results of three image bootstrapping strategies on LVLMs. For each benchmark, 'Vanilla' represents the original results, while $\mathcal{V}_1$, $\mathcal{V}_2$, and $\mathcal{V}_3$ represent the results after image bootstrapping. (The numbers in parentheses show the difference compared to vanilla, with red indicating decreasing, and green increasing.)

| Model | SEEDBench (%) | | | | MMBench (%) | | | | MME (%) | | | |
|---|---|---|---|---|---|---|---|---|---|---|---|---|
| | Vanilla | $\mathcal{V}1$ | $\mathcal{V}2$ | $\mathcal{V}3$ | Vanilla | $\mathcal{V}1$ | $\mathcal{V}2$ | $\mathcal{V}3$ | Vanilla | $\mathcal{V}1$ | $\mathcal{V}2$ | $\mathcal{V}3$ |
| DeepSeek-VL | 69.44 | 64.79(4.65↓) | 70.35(0.91↑) | 69.39(0.05↓) | 79.72 | 75.90(3.82↓) | 80.03(0.31↑) | 77.96(1.76↓) | 86.08 | 78.16(7.92↓) | 86.59(0.51↑) | 82.69(3.39↓) |
| TransCore-M | 73.58 | 69.10(4.48↓) | 73.86(0.28↑) | 69.52(4.06↓) | 79.64 | 75.37(4.27↓) | 79.72(0.08↑) | 76.89(2.75↓) | 88.19 | 83.13(5.06↓) | 88.87(0.68↑) | 86.15(2.03↓) |
| Monkey-Chat | 69.58 | 65.04(4.54↓) | 70.92(1.34↑) | 68.04(1.54↓) | 80.26 | 75.37(4.89↓) | 80.29(0.03↑) | 79.11(1.15↓) | 86.34 | 79.59(6.75↓) | 87.09(0.75↑) | 84.03(2.31↓) |
| LLaVA-NeXT-Vicuna | 71.54 | 65.60(5.94↓) | 71.88(0.34↑) | 67.83(3.71↓) | 79.26 | 74.04(5.22↓) | 79.27(0.01↑) | 77.74(1.52↓) | 68.97 | 59.61(9.36↓) | 69.39(0.42↑) | 68.46(0.51↓) |
| Qwen-VL-Chat | 69.58 | 65.04(4.54↓) | 70.92(1.34↑) | 68.04(1.54↓) | 71.84 | 67.31(4.53↓) | 72.37(0.53↑) | 72.69(0.85↓) | 74.36 | 67.03(7.32↓) | 78.16(3.79↑) | 81.73(7.37↑) |
| XComposer2 | 75.40 | 69.18(6.22↓) | 75.19(0.21↓) | 71.13(4.27↓) | 84.62 | 77.23(7.39↓) | 85.19(0.57↑) | 76.89(2.75↓) | 92.32 | 89.20(3.11↓) | 93.25(0.93↑) | 94.03(1.71↑) |
| Yi-VL-34B | 68.25 | 62.84(5.41↓) | 67.69(0.56↓) | 65.24(3.01↓) | 81.63 | 74.49(6.83↓) | 81.46(0.17↓) | 79.57(2.06↓) | 80.86 | 73.25(7.60↓) | 82.63(1.76↑) | 76.53(4.32↓) |
| InternVL-2 | 76.80 | 70.46(6.34↓) | 77.67(0.87↑) | 74.45(2.34↓) | 88.59 | 80.33(8.26↓) | 89.28(0.68↑) | 83.01(5.57↓) | 80.01 | 72.70(7.31↓) | 82.31(2.29↑) | 76.53(3.48↓) |
| GPT-4o | 77.59 | 70.91(6.68↓) | 78.11(0.52↑) | 73.70(3.89↓) | 87.37 | 82.28(6.68↓) | 89.06(1.68↑) | 85.28(2.09↓) | 77.57 | 71.04(6.52↓) | 78.49(0.92↑) | 72.30(5.26↓) |

Table 2: Results of vanilla and applying language bootstrapping strategy $\mathcal{L}_1$, $\mathcal{L}_2$, $\mathcal{L}_3$, $\mathcal{L}_4$ on LVLMs.

| Model | SEEDBench (%) | | | | | MMBench (%) | | | | | MME (%) | | | | |
|---|---|---|---|---|---|---|---|---|---|---|---|---|---|---|---|
| | Vanilla | $\mathcal{L}1$ | $\mathcal{L}2$ | $\mathcal{L}3$ | $\mathcal{L}4$ | Vanilla | $\mathcal{L}1$ | $\mathcal{L}2$ | $\mathcal{L}3$ | $\mathcal{L}4$ | Vanilla | $\mathcal{L}1$ | $\mathcal{L}2$ | $\mathcal{L}3$ | $\mathcal{L}4$ |
| DeepSeek-VL | 69.44 | 68.67(0.77↓) | 68.51(0.93↓) | 71.17(1.73↑) | 70.16(0.72↑) | 79.72 | 78.15(1.57↓) | 78.92(0.80↓) | 84.26(4.54↑) | 79.07(0.65↓) | 86.08 | 85.67(0.41↓) | 84.05(2.03↓) | 92.23(6.25↑) | 71.18(14.90↓) |
| TransCore_M | 73.58 | 72.94(0.64↓) | 71.67(1.91↓) | 71.95(1.63↓) | 72.19(1.39↓) | 79.64 | 78.53(1.11↓) | 78.62(1.02↓) | 84.89(5.25↑) | 78.55(1.09↓) | 88.19 | 85.57(2.62↓) | 86.31(1.88↓) | 87.42(0.77↓) | 72.92(15.27↓) |
| Monkey-Chat | 69.58 | 67.41(2.17↓) | 69.16(0.42↓) | 70.60(1.02↑) | 67.95(1.63↓) | 80.26 | 78.59(1.67↓) | 79.41(0.85↓) | 83.63(3.37↑) | 78.00(2.26↓) | 86.34 | 89.86(3.52↑) | 83.97(2.37↓) | 95.15(8.81↑) | 79.43(6.91↓) |
| LLaVA-NeXT-Vicuna | 71.54 | 70.16(1.38↓) | 69.86(1.68↓) | 73.06(1.52↑) | 70.61(0.93↓) | 79.26 | 78.84(0.42↓) | 77.90(1.36↓) | 83.80(4.54↑) | 77.14(2.12↓) | 68.97 | 65.73(3.24↓) | 69.97(1.00↑) | 80.49(11.52↑) | 63.43(5.54↓) |
| Qwen-VL-Chat | 63.62 | 60.01(3.61↓) | 60.84(2.78↓) | 61.05(2.57↓) | 52.98(10.64↓) | 71.84 | 68.29(3.55↓) | 69.73(2.11↓) | 69.54(2.30↓) | 61.32(10.52↓) | 92.32 | 39.28(35.08↓) | 63.51(10.85↓) | 79.21(4.85↑) | 54.18(20.18↓) |
| XComposer2 | 75.40 | 73.89(1.51↓) | 73.38(2.02↓) | 75.90(0.50↑) | 74.34(1.06↓) | 84.62 | 83.80(0.82↓) | 84.04(0.58↓) | 88.57(3.95↑) | 83.64(0.98↓) | 92.32 | 88.35(3.97↓) | 91.12(1.20↓) | 92.45(0.13↑) | 73.74(18.58↓) |
| Yi-VL-34B | 68.25 | 66.34(1.91↓) | 66.30(1.95↓) | 71.49(3.24↑) | 66.70(1.55↓) | 81.63 | 80.39(1.24↓) | 80.66(0.97↓) | 86.39(4.76↑) | 78.96(2.67↓) | 80.86 | 82.74(1.88↑) | 79.66(1.20↓) | 89.21(8.35↑) | 65.09(15.77↓) |
| InternVL-2 | 76.80 | 75.26(1.53↓) | 74.77(2.03↓) | 77.24(0.43↑) | 74.31(2.48↓) | 88.59 | 85.38(3.21↓) | 87.52(1.07↓) | 89.59(1.00↑) | 85.53(3.06↓) | 80.01 | 77.15(2.85↓) | 79.34(0.67↓) | 79.52(0.49↓) | 69.98(10.03↓) |
| GPT-4o | 77.59 | 77.01(0.57↓) | 75.89(1.70↓) | 78.29(0.70↑) | 74.69(2.90↓) | 87.37 | 86.61(0.76↓) | 85.84(1.53↓) | 87.60(0.99↑) | 81.10(5.51↓) | 77.57 | 76.22(1.34↓) | 70.99(6.57↓) | 78.03(0.46↑) | 73.35(4.22↓) |

**Evaluated LVLMs.** Our evaluated LVLMs include both closed-source APIs: GPT-4o (OpenAI, 2024) and Claude (Anthropic, 2024), and open-source models: DeepSeek-VL (Lu et al., 2024b), TransCore_M (Research, 2023), XComposer2 (Dong et al., 2024), Monkey-Chat (Li et al., 2024b), LLaVA-NeXT-Vicuna (Liu et al., 2024a), Yi-VL-34B (Young et al., 2024), Qwen-VL (Bai et al., 2023) and InternVL (Chen et al., 2023). We utilize the standardized evaluation platform VLMEvalkit (Duan et al., 2024) and set the generation temperature as 0 for all evaluated LVLMs to ensure a fair comparison.

## 5.2 RESULT OF SINGLE BOOTSTRAPPING STRATEGY

**The results of image bootstrapping strategies.** We apply our designed image bootstrapping strategies: Adding New Objects ($\mathcal{V}1$), Remove Existing Objects ($\mathcal{V}2$), and Expanding Original Images ($\mathcal{V}3$) on existing benchmarks. Table 1 presents the evaluation results across a series of LVLMs. Consistent with predefined difficulty levels, $\mathcal{V}_1$ and $\mathcal{V}_3$ generally result in a decrease in accuracy, and $\mathcal{V}2$ slightly boost the performance. It indicates that adding irrelevant objects or expanding the original image can introduce more superfluous visual content, thereby hindering LVLMs from capturing the key visual content related to the problem. Meanwhile, removing some irrelevant, extraneous objects can simplify the visual information of the image, allowing LVLMs to better focus on the key elements. More visualized cases are in Appendix A.3 and A.4.

**The results of language bootstrapping strategies.** We also apply linguistic bootstrapping on different linguistic levels, including word substitution ($\mathcal{L}1$), sentence rephrasing ($\mathcal{L}2$), relevant context($\mathcal{L}3$), and irrelevant context ($\mathcal{L}4$). Table 2 illustrates that, except for $\mathcal{L}_3$, the other strategies result in a degradation of LVLM performance. This highlights the challenge LVLMs face in addressing questions posed by different individuals from diverse backgrounds. Therefore, further efforts are needed to enhance LVLMs' ability to comprehend the semantic essence of questions and improve their user adaptability. These findings align with our previous revisiting experiments, suggesting that benchmarks may be contaminated. Conversely, incorporating relevant captions $\mathcal{L}3$ helps the model better understand images, boosting accuracy. Quantitatively, employing $\mathcal{L}3$ and $\mathcal{L}4$ presents more fluctuation than $\mathcal{L}1$ and $\mathcal{L}2$. This may be due to the situation (Kamradt, 2024) that longer text poses a greater influence for LVLMs compared to shorter words and sentence.

## 5.3 RESULT OF COMPOSITIONAL BOOTSTRAPPING STRATEGY

**The performance of paired multimodal composition.** We combine our strategies on $\mathcal{V}$ and $\mathcal{L}$ and obtained 12 dynamic variants that integrate visual and linguistic bootstrapping. Figure 5 shows the composition results in MMBench across three LVLMs, where each axis is organized from easy to difficult based on the complexity levels obtained from the above single-strategy experiments. We

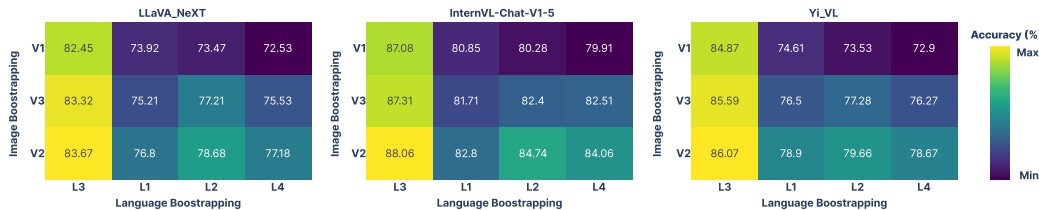

Figure 5: Results of composing image and language bootstrapping strategies on MMBench.

Table 3: Results of the hardest and easiest variants after multimodal bootstrapping strategy composition. The 'vanilla' represents the original results, while the hardest is $\mathcal{V}_1 + \mathcal{L}_4$, easiest is $\mathcal{V}_2 + \mathcal{L}_3$.

| Model | MMBench_DEV_Full (%) | | | MMvet | | | LLaVABench | | |
|---|---|---|---|---|---|---|---|---|---|
| | Hard ($\mathcal{V}1+\mathcal{L}4$) | vanilla | Easy ($\mathcal{V}2+\mathcal{L}3$) | Hard ($\mathcal{V}1+\mathcal{L}4$) | vanilla | Easy ($\mathcal{V}2+\mathcal{L}3$) | Hard ($\mathcal{V}1+\mathcal{L}4$) | vanilla | Easy ($\mathcal{V}2+\mathcal{L}3$) |
| DeepSeek-VL | 73.82 | 78.30 | 83.06 | 22.88 | 41.88 | 64.03 | 40.20 | 58.80 | 66.00 |
| Monkey-Chat | 73.06 | 77.45 | 82.65 | 29.72 | 40.96 | 40.96 | 38.70 | 48.50 | 73.00 |
| LLaVA-NeXT-Vicuna | 71.00 | 76.41 | 82.83 | 24.22 | 47.56 | 63.02 | 35.50 | 55.20 | 69.00 |
| XComposer2 | 76.94 | 83.02 | 87.08 | 30.55 | 38.48 | 60.55 | 42.30 | 58.00 | 73.50 |
| Yi-VL-34B | 71.40 | 77.91 | 85.30 | 22.15 | 30.55 | 60.22 | 37.50 | 44.30 | 59.70 |
| InternVL-2 | 78.00 | 86.78 | 88.17 | 36.33 | 54.49 | 68.15 | 36.00 | 53.00 | 68.30 |
| GPT-4o | 79.55 | 85.27 | 86.32 | 43.66 | 73.94 | 77.15 | 55.20 | 76.00 | 78.80 |
| Claude3-5V-Sonnet | 69.04 | 82.36 | 85.17 | 26.10 | 70.77 | 75.91 | 36.80 | 70.50 | 75.8 |

observe that these LVLMs exhibit varying performance when faced with different variants, with overall trends being relatively consistent (*i.e.*, performance gradually decreases from the lower left to the upper right corner). Specifically, the combination of $\mathcal{V}1 + \mathcal{L}4$ presents hardest challenges for LVLMs, while $\mathcal{V}2 + \mathcal{L}3$ significantly boosts performance. These findings align with our observations from applying single-modal strategies. Results of paired multimodal composition on more benchmarks and LVLMs can be seen in Appendix figure 12.

Based on the above result, we extend these two extreme compositions (*i.e.,* $\mathcal{V}1 + \mathcal{L}4$ and $\mathcal{V}2 + \mathcal{L}3$ ) to the full set of MMBench, MMvet, and LLaVABench, to demonstrate the universality of our dynamic evaluation strategies. As displayed in Table 3, our VLB could be effective in evaluating the upper and lower bounds for the models' performance. This capability to transform an original sample into versions with varying complexity not only tests but also enhances our understanding of the comprehensive capabilities of LVLMs. Through VLB, we achieve a more thorough evaluation, highlighting the robustness and adaptability of LVLMs to handle diverse and complex multimodal interactions.

**The performance of multi-strategy composition.** Due to every single VLB strategy for image and question being atomic, we can explore the effect of different compositions according to the number of applied strategy: vanilla, single-strategy ($\mathcal{V}_1$), dual-strategy($\mathcal{V}_1+\mathcal{V}_3$), and tri-strategy ($\mathcal{V}_1+\mathcal{V}_3+\mathcal{L}_4$). Together with the easiest composition ($\mathcal{V}_2+\mathcal{L}_3$), Figure 6 demonstrates the performance of three advanced LVLMs on SEEDBench and MMvet. The lines show that as the number of hard strategies we use increases from 0-3, the difficulty of the benchmark also increases, leading to a gradual decline in the accuracy of LVLM.

### 5.4 ABLATION STUDY AND ANALYSIS

**The effect of outpainting ratio in $\mathcal{V}_3$.** Since different expanding ratios can result in varied visual content, we investigated the influence of different expanding ratios ($1.25\times$, $1.50\times$, $1.75\times$, and $2\times$ ) in our $\mathcal{V}3$ strategy. The results in Figure 7 indicate that as the expanding ratios increase, the accuracy of the models in correctly answering questions decreases, and the rate of this decrease becomes steeper. This can be attributed to the fact that a larger expanding ratio introduces a more prominent and possibly distracting background into the image, whose visual information may not be directly relevant to the question, thereby diverting the LVLM's attention from the crucial cues.

**Can our VLB reduce data contamination?** Our dynamic strategy mitigates data contamination in two ways. Firstly, since the new images and texts are generated using image editing tools and GPT-4V, where the process is random and variable, it is difficult for models to take the vast array of random variants into training. Secondly, the newly generated sample differs from the contaminated original sample, thereby reducing the similarity with training data. We applied the same methods

of Section 3 to redetect the hardest variant in Figure 8 and found a significant reduction in data contamination rate among training sets.

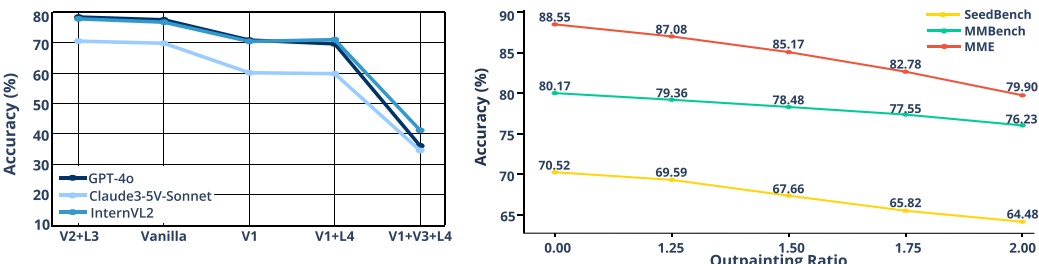

Figure 6: The accuracy of LVLMs towards different multi-strategy compositions on SEEDBench.

Figure 7: The average accuracy of all evaluated LVLMs on different ratios of outpainting scales.

**The effect of VLB on different tasks.** Since our dynamic strategies are applicable to a wide range of multimodal tasks, we utilize SEEDBench to demonstrate the performance changes of LVLMs across various tasks. Figure 9 presents a comparison of average performance among the vanilla benchmarks, and two dynamic variants with $\mathcal{V}2 + \mathcal{L}3$ and $\mathcal{V}1 + \mathcal{L}4$ strategies. It is evident that performance changes more significantly in tasks like 'Instance Interaction', 'Text Understanding' and 'Spatial Relation'. We found that images in 'Instance Interaction' and 'Spatial Relation' contain fewer objects, making the addition of new objects particularly disruptive to the LVLMs' focus on core visual elements. For 'Text Understanding', it aligns with our strategic design hypothesis that introducing noise contexts can distract the models from understanding essential texts. Detailed error cases of these tasks on GPT-4o are in Appendix A.6.

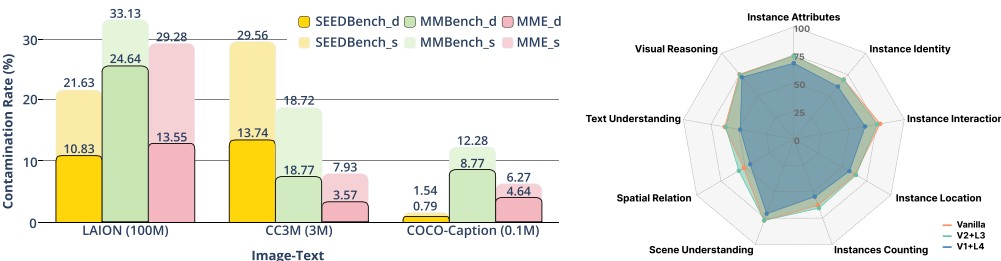

Figure 8: Reduction of image-text data contamination rate between the dynamic variant $d$ and the vanilla static benchmark $s$.

Figure 9: The accuracy for the hard variant, easy variant, and the vanilla static benchmark across different multimodal tasks on SEEDBench.

## 6 CONCLUSION AND DISCUSSION

In this paper, we begin by quantitatively detecting the degree of contamination present in current benchmarks, indicating the need for dynamic multimodal evaluation (DME). And then we develop the first DME protocol with flexible complexity via Vision-Language Bootstrapping (VLB). VLB introduces bootstrapping strategies designed to transform original images and questions into new versions that simulate real user interactions in LVLMs, focusing on aspects of visual attention and linguistic understanding. We also employ predefined rules and the judge to ensure the newly generated problem preserves the consistency with the vanilla. With VLB, we build a series of dynamic variants upon an existing benchmark with diverse complexities. By conducting extensive evaluations using VLB, we demonstrate the limitations of current LVLMs, highlighting areas for potential improvement of LVLM in user interaction and adaption.

## 7 ACKNOWLEDGMENTS

This paper is partially supported by the National Key R&D Program of China (NO.2022ZD0160102, NO.2022ZD0161000), and the General Research Fund of Hong Kong No.17200622 and 17209324. This work was done during his internship at Shanghai Artificial Intelligence Laboratory.

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

# A APPENDIX

## A.1 BOOTSTRAPPING STRATEGY INSTRUCTION

In this section, we detail the instructions for GPT-4 during the bootstrapping strategy process in Table 4. They are strategy $\mathcal{V}_1$, $\mathcal{V}_2$, $\mathcal{L}_3$ and $\mathcal{L}_4$, respectively.

In strategy $\mathcal{V}_1$, we designed add-instruction, to prompt GPT-4V to return a set of insertable objects. Each object is in the form of (object name, bounding box).

Since GPT-4V is relatively weak in fine-grained visual grounding, in strategy $\mathcal{V}_2$, we utilize semantic masks to facilitate the appropriate selection of objects for removal. Leveraging segmentation models, we obtain semantic masks $\mathbf{M}$ of all objects within an image. Upon identifying semantic masks $\mathbf{M}$, each object is assigned a serial number. GPT-4V requires to pinoint the objects suitable for removal based on their numbers and names, each formatted as (object name, serial number).

In strategy $\mathcal{L}_3$, we seek to enhance the context with a caption focusing on information related to the question, generated by GPT-4V. Thus, we query GPT-4V to generate additional information that aids in responding.

In strategy $\mathcal{L}_4$, we aims at introducing nosing context, which pertains to adding context that is related to the image but not directly pertinent to the question. For instance, an observer might initially notice another object or the overall atmosphere before focusing on a specific object about which the question is posed.

| Aspect | Input | Instructions |
|---|---|---|
| $\mathcal{V}_1(I)$ Image Bootstrapping | $<E_s>$ | There is a question about the image (resolution:$H \times W$) Its question is: $<Q>$. Its correct answer choice is: $<A>$. Now please add an object into this image, but keep the original answer of the question the same, which means the added object can not change the original answer and the position of the added object can not cover the visual answer information. Note that the size of the added object should be larger. Please give me randomly 10 objects can be added and give me the exact bounding box coordinates of each object according to the original resolution I give you. The box in the output is the coordinates of the upper left corner $(x_{min}, y_{min})$ and lower right corner $(x_{max}, y_{max})$ of the object. The output format should only be a list. Each item in the list must be a dict. Do not output any extra information! Just output a list. The example of output format is: $[\text{name} : \text{name of object1}, \text{box} : [x_{min}, y_{min}, x_{max}, y_{max}], \text{name} : \text{name of object2}, \text{box} : [x_{min}, y_{min}, x_{max}, y_{max}], \cdots]$. |
| $\mathcal{V}_2(I)$ Image Bootstrapping | $<E_s, M>$ | Based on the image I provided (image includes the mark and the mask for each object), tell me which objects can be removed without changing the answer to the question: $<Q>$. Please give me a list containing exactly 5 objects can be removed. Do not output any extra information. Just output a list. The example of output format is: $[\text{object\_mark} : \text{xxx}, \text{object\_name} : \text{xxx}, \text{object\_mark} : \text{xxx}, \text{object\_name} : \text{xxx}, \cdots]$ where object_mark is the given mark of object, and object_name is the name of specific object(can be repeated without distinction). Note that the removed object can not change the answer to the question. In the order you think is most obvious and appropriate to remove. |
| $\mathcal{L}_3(I)$ Text Bootstrapping | $<I, Q>$ | Please add context to the question: introducing context or details to the question that are relevant but not directly helpful in answering it. Refer to the accompanying image $$ for context. Question: $<Q>$. |
| $\mathcal{L}_4(I)$ Text Bootstrapping | $<I, Q>$ | Now, based on this image, I want you to provide a paragraph that is irrelevant to the question: $<Q>$. The paragraph should not help in answering the question at all. Be careful not to directly describe the image, as that would aid in answering the question. Instead, you should focus on an unrelated object or element in the image $$, and let your imagination run wild. The more creative and off-topic, the better. Please note the paragraph should be no more than 100 words. |

Table 4: The detailed instructions for image and language bootstrapping strategies.

## A.2 JUDGE INSTRUCTION

Although our dynamic pipeline adheres to strict criteria to maintain the consistency of the original intent and informational content, it still unintentionally alters the meaning. Therefore, we introduce the judge module to compare the original with the modified images and the original with the rephrased questions, ensuring the preservation of the essence and factual accuracy. Specifically, to minimize excessive perturbations, the judge module assesses the newly generated images and questions from both the visual and textual modalities, respectively. Details of instructions can be seen in the following Table 5.

## A.3 MORE CASES OF IMAGE BOOTSTRAPPING

We provide some cases transformed by our image bootstrapping strategies in Figure 10. Visually, these dynamic variants show consistency with original images, ensuring the preservation of the

| Aspect | Input | Instructions | Output |
|---|---|---|---|
| Image Judgement | $< E_s, \mathcal{V}_1(I) >$ | Image-1: $< I >$ Image-2: $< \mathcal{V}(I) >$. I added an object to the image carefully. Please assess whether the modifications in the images affect the answer to the corresponding question. You will see two images: one is the original image (Image-1), and the other is the modified image (Image-2). The question is as follows: $< Q >$ with its associated answer: $< A >$. Please carefully compare the two images and judge whether the changes in the modified image (Image-2) would affect the correctness of the answer to the question relative to the original image (Image-1). Answer with "Yes" or "No" only. | 'Yes' / 'No' |
| Image Judgement | $< E_s, \mathcal{V}_2(I) >$ | Image-1: $< I >$ Image-2: $< \mathcal{V}(I) >$. I removed an object from the image carefully. Please assess whether the modifications in the images affect the answer to the corresponding question. You will see two images: one is the original image (Image-1), and the other is the modified image (Image-2). The question is as follows: $< Q >$ with its associated answer: $< A >$. Please carefully compare the two images and judge whether the changes in the modified image (Image-2) would affect the correctness of the answer to the question relative to the original image (Image-1). Answer with "Yes" or "No" only. | 'Yes' / 'No' |
| Image Judgement | $< E_s, \mathcal{V}_3(I) >$ | Image-1: $< I >$ Image-2: $< \mathcal{V}(I) >$. I expanded the image carefully. Please assess whether the modifications in the images affect the answer to the corresponding question. You will see two images: one is the original image (Image-1), and the other is the modified image (Image-2). The question is as follows: $< Q >$ with its associated answer: $< A >$. Please carefully compare the two images and judge whether the changes in the modified image (Image-2) would affect the correctness of the answer to the question relative to the original image (Image-1). Answer with "Yes" or "No" only. | 'Yes' / 'No' |
| Text Judgement | $< E_s, \mathcal{L}_1(Q) >$ | The two provided questions are semantically same and only have some minor differences. Question1: $< Q >$ Question2: $< \mathcal{L}_1(Q) >$. The Ground Truth answer is: $< A >$. Please determine whether the ground truth answer applies to both Question 1 and Question 2? Only provide a simple output: Yes or No. | 'Yes' / 'No' |
| Text Judgement | $< E_s, \mathcal{L}_2(Q) >$ | Image: $< I >$ Given an image along with its corresponding question and answer, the original question has been modified into a semantically similar one. The original question is $< Q >$, with its associated answer: $< A >$, and the modified question is $< \mathcal{L}_2(Q) >$. Please determine whether the modified question changes the semantics of the original question, thereby potentially affecting the correctness of the original answer. If it does change, output "Yes"; if it does not change, output "No". Please provide a simple output without explanation. | 'Yes' / 'No' |
| Text Judgement | $< E_s, \mathcal{L}_3(Q) >$ | Image: $< I >$ There is a question-answer pair regarding this image. Under the premise of not changing the semantics of the question, I added some image descriptions into the original question, and got a modified question. The original question is: $< Q >$, the modified question is: $\mathcal{L}_3(Q) >$ and the original answer is: $< A >$. Based on these information, do the modified question and the original question ask the same thing? Namely, are they both asking $< Q >$? If they are, output "Yes" and if they are not asking the same thing, output "No". Only provide a simple output without explanation. | 'Yes' / 'No' |
| Text Judgement | $< E_s, \mathcal{L}_4(Q)) >$ | Image: $< I >$ There is a question-answer pair regarding this image. Under the premise of not changing the semantics of the question, I added some image descriptions into the original question, and got a modified question. The original question is: $< Q >$, the modified question is: $\mathcal{L}_3(Q) >$ and the original answer is: $< A >$. Based on these information, do the modified question and the original question ask the same thing? Namely, are they both asking $< Q >$? If they are, output "Yes" and if they are not asking the same thing, output "No". Only provide me with a simple output without explanation. | 'Yes' / 'No' |

Table 5: The detailed instruction given to the judge module.

essence. However, for LVLMs, the scenario is quite different. Despite the visual similarity, these modified images present new challenges.

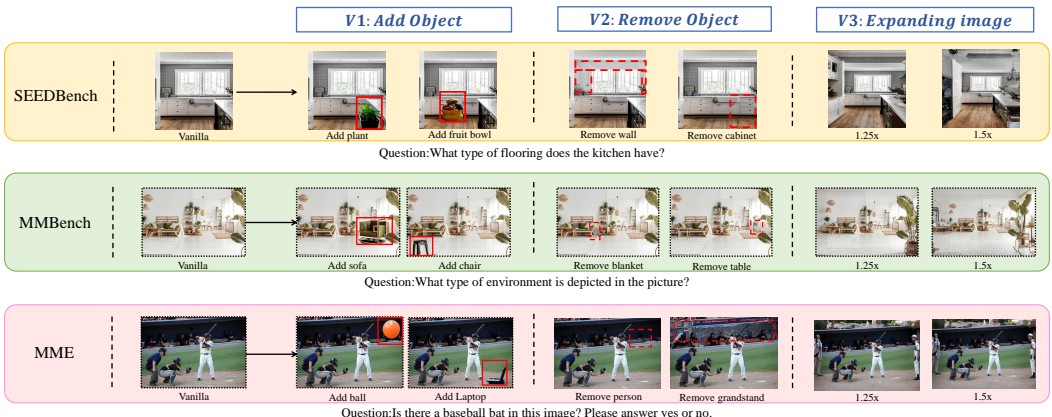

Figure 10: More cases of image bootstrapping on SEEDBench, MMBench, and MME

## A.4 MORE CASES OF LANGUAGE BOOTSTRAPPING

We present some results of bootstrapping the textual modality of existing benchmarks in Table 6.

## A.5 THE EFFECTIVENESS OF JUDGE MODULE.

Although our newly transformed images and questions have passed an adversarial judged, for the sake of rigor, we further conduct a human verification to ensure that our variants maintain consistent with the original benchmark both visually and linguistically. For each dynamic strategy, we recruit 20 human experts (with bachelor or higher degree) and randomly selected 100 samples from each strategy for each benchmark, totally 2100 samples. They are also tasked with judging the integrity and appropriateness of the modifications. For visual dynamic strategy, we compose the original and edited image, with the corresponding question and answer, into the questionnaire. The experts are required to verify: (1) Whether the edited image maintains the same answer as the original image. For linguistics dynamic strategy, we compose the original and transformed question, with the corresponding image and answer, into the questionnaire. And ask (2) whether the original and paraphrased questions were equivalent. The evaluation required a simple 'Yes' or 'No' answer for our asked questions (1) and (2).

Experts independently reviewed samples. Once all experts completed their evaluations, a voting process ensued. A sample was only considered to have passed if more than half of the experts

| Benchmark | Strategy | Question |
|---|---|---|
| SEEDBench | Original | What type of flooring does the kitchen have? |
| | $\mathcal{L}_1(Q)$ | What type of flooring does the cookroom have? |
| | $\mathcal{L}_2(Q)$ | The kitchen is equipped with what kind of flooring? |
| | $\mathcal{L}_3(Q)$ | A stylish and modern kitchen interior design. There is a large window above the sink that allows ample natural light to flood the space. The countertops appear to be a solid surface, possibly marble or quartz. What type of flooring does the kitchen have? |
| | $\mathcal{L}_4(Q)$ | On the kitchen cupboard, there is a cookbook inviting people to a baking challenge that spreads the joie de vivre of pies. This cookbook transforms an everyday kitchen into a nexus of community, where the artful dance of flour and sugar ignites a passion for the pastry arts amongst friends. What type of flooring does the kitchen have? |
| MMBench | Original | What type of environment is depicted in the picture? |
| | $\mathcal{L}_1(Q)$ | What type of surroundings is depicted in the picture? |
| | $\mathcal{L}_2(Q)$ | What sort of setting does the picture portray? |
| | $\mathcal{L}_3(Q)$ | The image portrays a modern, minimalist living space featuring an open living and dining area. The room incorporates a neutral color palette with soft earthy tones and ample natural light. The furniture and decor are a blend of natural materials like wood and woven fibers, contributing to a cozy and clean aesthetic. The presence of several houseplants brings a touch of nature indoors, emphasizing a fresh and inviting atmosphere. The environment is carefully organized and designed to create a harmonious and tranquil living space. Given the details in the image, the depicted environment can best be described as a modern, minimalist, and cozy indoor living space. What type of environment is depicted in the picture? |
| | $\mathcal{L}_4(Q)$ | Imagine if instead of inorganic matter, all furniture was bioengineered, possessing a life cycle akin to plants or animals. Consider a sofa that sprouts small, cushiony shoots each spring, which over the summer gradually bloom into lush, velvety surfaces perfect for lounging. As autumn approaches, the colors of the cushions would transform into a spectacular display of reds, golds, and oranges, before gently shedding layers, not dissimilar to deciduous trees. The cycle would then culminate in a period of dormancy during winter, when the sofa regathers its strength to ensure it can unfurl its comfy resplendence when warmth returns. What type of environment is depicted in the picture? |
| MME | Original | Is there a baseball bat in this image? Please answer yes or no. |
| | $\mathcal{L}_1(Q)$ | Is there a baseball bat in this figure? Please answer yes or no. |
| | $\mathcal{L}_2(Q)$ | Does there exist baseball bat in this image? Please give me an answer just yes or no. |
| | $\mathcal{L}_3(Q)$ | The image depicts a scene from a baseball game. The batter is standing in the batter's box, poised to swing, while the catcher is crouched behind the home plate, ready to catch the ball. An umpire stands behind the catcher, observing the scene closely. The dugout in the background is populated with team members watching the game. Question: Is there a baseball bat in this image? Please answer yes or no. |
| | $\mathcal{L}_4(Q)$ | In the world of sporting attire, the evolution of the catcher's mitt is particularly fascinating. This pivotal piece of equipment, cradled in traditions as old as the game itself, has undergone a considerable transformation over the years. Initially just leather pads to protect the hands, modern catcher's mitts are now engineered marvels featuring specialized cushioning, tailored fit, and durable materials that withstand repetitive high-impact catches. These mitts not only protect the player's hands but also enhance their ability to catch fast-moving balls with precision, showcasing a perfect blend of heritage and innovation in sports technology. Question: Is there a baseball bat in this image? Please answer yes or no. |

Table 6: More cases of language bootstrapping on SEEDBench, MMBench, and MME.

deemed it consistent with the original. In cases where the voting resulted in a tie, the sample was regarded as ambiguous and deemed not to maintain consistency with the original. The positive results are shown in Figure 11, our human verification obtain a high accuracy rate, average of 96% for image dynamic strategy and 97% for question dynamic strategy on all three datasets. The result showcases a high level of alignment between the judge model and human verification, proving the equivalence and correctness of our dynamic methodology.

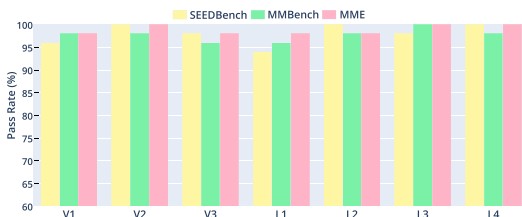

Figure 11: Human verification rate on each bootstrapping variant maintaining consistency with the vanilla.

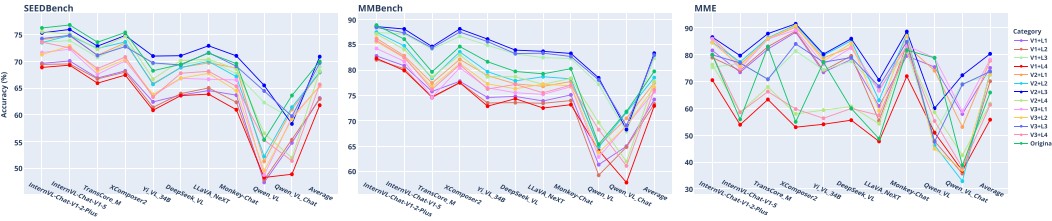

Figure 12: Results of composition between image bootstrapping $\mathcal{V}$ and language bootstrapping $\mathcal{L}$ strategy.

## A.6 ANALYSIS OF ERROR CASE

We visualize the GPT4o's responses to images before and after dynamic changes. Figure 13(a) shows an example of instance identity task, where adding a sofa caused the LVLM to wrongly answer a question it correctly responded in the original. This may be due to the added red sofa, which is brightly colored and distracts the model's attention. Figure 13 (b) is an example of spatial relation task, where the removal of a cabinet made the relative positions of the fan and stage clearer, allowing the LVLM to correctly answer the question it had previously gotten wrong.

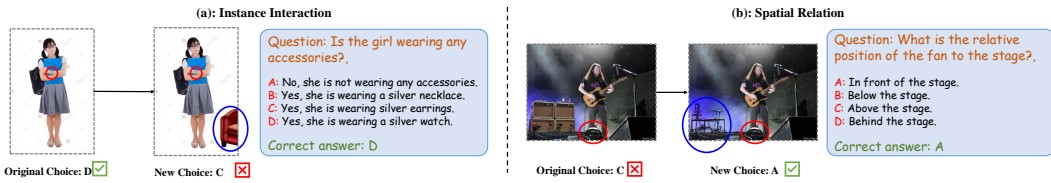

Figure 13: (a) After the $\mathcal{V}_1$ strategy, GPT-4 incorrectly answered an instance identification question it had previously answered correctly. (b) By $\mathcal{V}_2$ strategy, GPT-4 correctly answered a spatial relation question it had previously answered incorrectly.

We further visualize more of LVLMs' responses in Figure 14, especially about questions before and after dynamic changes. As can be seen, our dynamic strategy for generating new multi-modal VQA pairs indeed results in performance changes, presenting a greater challenge for LVLMs.

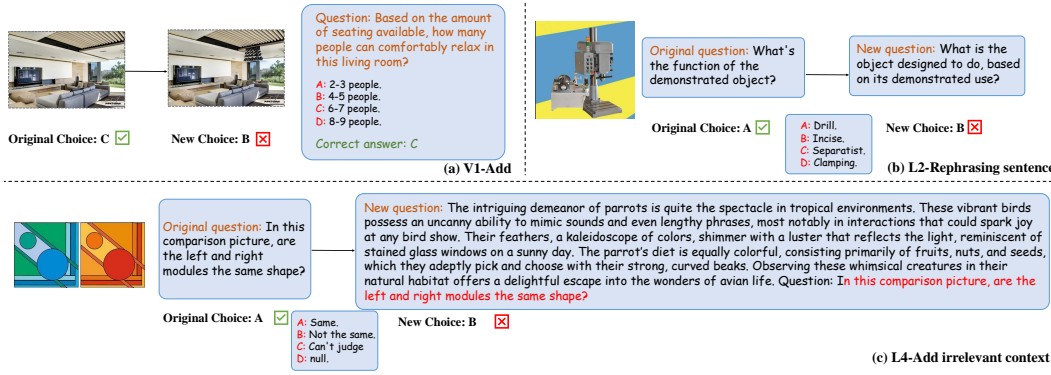

Figure 14: More error cases in more LVLMs and more strategies. (a) After the $\mathcal{V}_1$ strategy, GPT-4 incorrectly answered a question it had previously answered correctly. (b) By $\mathcal{L}_2$ strategy, InternVL incorrectly answered a rephrased question it had previously answered correctly. (c) By $\mathcal{L}_4$ strategy, deepseek-vl incorrectly answered a question with irrelevant context.

## A.7 EXPERIMENTAL DETAIL

### A.7.1 THE SPECIFIC VLMS PARAM VERSION

Table 7 lists the specific model names and their parameter version evaluated in our experiment.

| Model | DeepSeek-VL | TransCore-M | Monkey-Chat | LLaVA-NeXT-Vicuna | Qwen-VL-Chat | XComposer2 | Yi-VL | InternVL |
|---|---|---|---|---|---|---|---|---|
| **Size** | 7B | 28B | 9.8B | 13B | 20B | 7B | 34B | 8B |

Table 7: Comparison of model sizes across different architectures.

## A.7.2 EXPERIMENT ON THE PERFORMANCE VARIANCE

In our practical experiment, for each image and language strategy, we set up five random seeds, generating five data variants. Therefore, all experimental results are the average of these five variants. Below we provide the standard deviation for each strategy across the five variants.

| Dataset | Model | V1 | V2 | V3 | L1 | L2 | L3 | L4 |
|---------|-------|-----|-----|-----|-----|-----|-----|-----|
| SEEDBench | TransCore-M | 69.10 (0.5737) | 73.86 (0.5026) | 69.52 (1.3760) | 72.94 (0.0353) | 71.67 (0.0330) | 71.95 (0.0599) | 72.19 (0.0319) |
| | DeepSeek | 64.79 (0.7716) | 70.35 (0.1560) | 69.39 (0.1825) | 68.67 (0.0523) | 68.51 (0.0475) | 71.17 (0.0655) | 70.16 (0.0447) |
| | InternVL2 | 70.46 (0.6834) | 77.67 (0.6368) | 74.45 (0.5221) | 75.26 (0.0486) | 74.77 (0.0487) | 77.24 (0.0630) | 74.31 (0.0444) |
| MMBench | TransCore-M | 75.37 (0.6665) | 79.72 (0.3891) | 76.89 (0.5022) | 78.53 (0.0056) | 78.62 (0.0032) | 84.89 (0.0050) | 78.55 (0.0016) |
| | DeepSeek | 75.90 (0.2780) | 80.03 (0.2573) | 77.96 (0.2327) | 78.15 (0.0037) | 78.92 (0.0025) | 84.26 (0.0042) | 79.07 (0.0044) |
| | InternVL2 | 80.33 (1.0671) | 89.28 (0.7612) | 83.01 (0.2827) | 85.38 (0.0032) | 87.52 (0.0015) | 89.59 (0.0038) | 85.53 (0.0049) |
| MME | TransCore-M | 83.13 (1.1170) | 88.87 (0.1922) | 86.15 (0.6643) | 85.57 (0.0298) | 86.31 (0.0522) | 87.42 (0.0534) | 72.92 (0.0689) |
| | DeepSeek | 78.16 (1.2455) | 86.59 (0.0841) | 82.69 (0.6797) | 85.67 (0.0425) | 84.05 (0.0181) | 92.23 (0.0478) | 71.18 (0.0603) |
| | InternVL2 | 72.70 (0.4262) | 82.31 (0.7943) | 76.54 (0.9789) | 77.15 (0.0277) | 79.34 (0.0474) | 79.52 (0.0563) | 69.98 (0.0455) |

Table 8: Mean and standard deviation (in parentheses) of accuracy across different strategies and datasets.

As can be seen from Table 8, the standard deviation in each strategy is slight, thus the variance scale caused by the randomness of GPT-4V and PowerPaint is also minimal. These demonstrate the reliability of the average accuracy result reported in our paper.

## A.8 ABLATION STUDY OF IMAGE EDITING MODEL

To explore our framework on other models with similar functions, we selected two other popular image editing models to replace PowerPaint. For strategy v1, we utilize BrushNet for the object addition step and keep all other steps exactly the same with PowerPaint. For strategy V3, we apply Stable Diffusion for image outpainting with the same ratio 1.5×. The table below shows the evaluation results of our framework with PowerPaint, BrushNet, and Stable Diffusion on SEEDBench across three LVLMs. It is worth noting that we also generated five variants for each setting and take their average results.

| Model | V1 | | | V3 | | |
|-------|----|----|----|----|----|----|
| | vanilla | +PowerPaint ($\downarrow$) | +Brushnet ($\downarrow$) | vanilla | +Powerpaint ($\downarrow$) | +SD outpainting ($\downarrow$) |
| TransCore-M | 73.58 | 69.10 (4.48$\downarrow$) | 70.91 (2.67$\downarrow$) | 73.58 | 69.52 (4.06$\downarrow$) | 70.45 (3.13$\downarrow$) |
| DeepSeek | 69.44 | 64.79 (4.65$\downarrow$) | 66.26 (3.18$\downarrow$) | 69.44 | 69.39 (0.05$\downarrow$) | 69.14 (0.30$\downarrow$) |
| InternVL-2 | 76.80 | 70.46 (6.34$\downarrow$) | 71.36 (5.44$\downarrow$) | 76.80 | 74.45 (2.34$\downarrow$) | 75.05 (1.75$\downarrow$) |

Table 9: Accuracy comparison of VLB under different image editing models for V1 and V3.

As can be seen from Table 9, although there are minor numerical differences between the results from PowerPaint and other editing models, their trends are similar. Therefore, the experimental conclusions in our original paper are valid and reliable. What's more, since our framework is verified to be adaptable to many tool models with similar functions, then our framework will perform better and better as these tool models are improving.

## A.9 ABLATION STUDY OF JUDGE MODULE

We further present the results of human verification on consistency before and after the judge module. In specific, we sampled data from generated images and questions that do not undergo adversarial selection by the judge module, and conducted the same human verification as A.5.

As shown in Table 10, it is evident that for most strategies, the verification rate is significantly improved after the judge module, demonstrating that our judge module is an effective automated verification technique. The increased approval rate for the visual strategy demonstrates the effective

Ju, Xuan, et al. "Brushnet: A plug-and-play image inpainting model with decomposed dual-branch diffusion." arXiv preprint arXiv:2403.06976 (2024).

Rombach, Robin, et al. "High-resolution image synthesis with latent diffusion models." Proceedings of the IEEE/CVF conference on computer vision and pattern recognition. 2022.

| | Dataset | V1 | V2 | V3 | L1 | L2 | L3 | L4 |
|---|---|---|---|---|---|---|---|---|
| | SEEDBench | 83 | 92 | 91 | 89 | 93 | 93 | 98 |
| Before Judge | MMBench | 81 | 94 | 88 | 91 | 90 | 94 | 95 |
| | MME | 84 | 95 | 87 | 87 | 85 | 95 | 97 |

Table 10: Human verification rate before judgment on different benchmarks and strategies.

judgment of our judge module in image-text alignment, while the improved approval rate for the linguistic strategy reflects the effective verification in terms of semantics.

| Dataset | V1 | V2 | V3 | L1 | L2 | L3 | L4 |
|---|---|---|---|---|---|---|---|
| SEEDBench | 0.3477 | 0.1014 | 0.2441 | 0.3428 | 0.0307 | 0.0303 | 0.0054 |
| MMBench | 0.4395 | 0.0906 | 0.2729 | 0.3092 | 0.0158 | 0.0867 | 0.0567 |
| MME | 0.4019 | 0.0973 | 0.2063 | 0.2085 | 0.3263 | 0.0548 | 0.1863 |

Table 11: The average adversarial iterations by our judge module for each strategy.

We have also provided the average adversarial iterations by our judge module for each strategy in Table 11. The iteration number means the average times of samples be rejected and re-selected(i.e., do not pass) by the judge model. This demonstrates that our judge module is an effective and intelligent filter, which can select accurately modified samples that are consistent with the original.

## A.10 MORE ANALYSIS ON EVALUATION RESULT

Visually, we analyzed the relationship between the ClipScore difference value and the accuracy difference value. For strategy V1, V2 and V3, we calculated the CLIPScore between newly generated images and the original images, together with the average accuracy difference of each strategy. Below we take MMbench as an example for analysis. As shown in the Table 12, it can be generally found that, images with a lower CLIPScore with the original image cause greater accuracy changes for LVLMs. This indicates that the relatively bigger modifications on the original image can lead to more significant visual attention disruptions and thus pose bigger challenges to the LVLMs.

| strategy | V1 | V3 | V2 |
|---|---|---|---|
| Accuracy Difference | 5.7655 | 2.2777 | 0.4133 |
| CLIPScore | 0.8150 | 0.8776 | 0.9502 |

Table 12: Relationship of accuracy difference and CLIPScore on MMBench. The table is ordered by the accuracy difference value compared to the original results for each strategy.

| strategy | L4 | L3 | L1 | L2 |
|---|---|---|---|---|
| Accuracy Difference | 3.2211 | 2.900 | 1.5944 | 1.1433 |
| Question Length | 85.7337 | 63.6534 | 8.8523 | 9.3083 |

Table 13: Relationship of accuracy difference and Question Length on MMBench.

Linguistically, for strategies L1, L2, L3, and L4, we calculated the word number of questions, namely question length. Then we analyzed the relationship between the question length and the accuracy differences, as depicted in the following Table 13. Generally, it can be observed that the longer question can lead to more accuracy changes for LVLMs. This also aligns with the conclusions in previous work, which reveals longer contents pose greater challenges to LVLMS. Therefore, the result of our linguistic strategy on LVLM is reasonable and systematic.

## A.11 DETAILED EXPLANATION OF USER INTERACTION

The user interaction with LVLM is defined by the cognitive process through which users comprehend the image and ask questions of their interest in the VQA setting. This process is influenced by visual attention and linguistic understanding.

Wang, Weiyun, et al. "Needle In A Multimodal Haystack." arXiv preprint arXiv:2406.07230 (2024).

Specifically, Visual attention affects the way how individuals selectively focus on specific visual elements within an image or scene while filtering out less relevant information. Since different users pose various identities and backgrounds, their levels of visual attention also vary. For instance, children's attention might be more easily distracted by irrelevant objects in images than educated adults. We categorize visual attention with concentration and distraction. The attention concentration is implemented by object removal while distraction is implemented by object addition and image outpainting.

Linguistic understanding indicates how individuals interpret and comprehend questions. This process influences the language proficiency of different users. Specifically, when different users utilize LVLMs, they may have different linguistic expressions towards a same question due to their distinct identities or educational backgrounds. Therefore, as shown in Figure 3(a), our work conducts transformation on the original question from three linguistic levels: word-level (L1), sentence-level (L2), and context-level (L3, L4).

**(1) Visually**, as described in section 4.2, since different users pose various identities and backgrounds, their levels of visual attention also vary. For instance, children's attention might be more easily distracted by irrelevant objects in images than educated adults. Therefore, as shown in Figure 3(a), we simulate the distraction, simplification, and expansion on visual attention for image bootstrapping, respectively corresponding to our three strategies : adding new objects(V1), removing existing objects(V2), and expanding original images(V3). Specifically in figure 3(c), compared to the original image, we added a plant, removed a wall, and outpainted the image with a ratio of 1.5, respectively, obtaining three images after visual dynamic strategies V1,V2,V3.

**(2) Linguistically**, as described in section 4.3, we also simulate the language usage on LVLMs from different users. Specifically, when different users utilize LVLMs, they may have different linguistic expressions towards a same question due to their distinct identities or educational backgrounds. Therefore, as shown in Figure 3(a), our work conducts transformations on the original question from three linguistic levels: word-level (L1), sentence-level (L2), and context-level (L3, L4) . For example in figure 3(c), the original question is 'What type of flooring does the kitchen have?'. From the word-level, some users might habitually use the word 'cookroom' instead of 'kitchen'. From the sentence level, more literary users like writers, might phrase it as 'is equipped with what kind of' instead of 'have what type of'. From the context level, more talkative users like teachers pretend to guide into the question by giving some scene description aforehead, which corresponds to our L3 strategy 'adding relevant context'. Meanwhile, children users, who may lack mature logical thinking ability, might first ramble about unrelated gadgets in the image before posing his realistic question to LVLMs. This corresponds to L4 strategy 'adding irrelevant context'.

Generally, our dynamic strategies effectively simulate various real user interaction scenarios from both visual and linguistic perspectives, which is very significant for the practical application and widespread adoption of LVLMs.

### A.12 MORE RESULTS ON DATA CONTAMINATION

| Pretraining-Dataset | Benchmark | Vanilla | L1 | L2 | L3 | L4 |
|---|---|---|---|---|---|---|
| **LAION (100M)** | SEEDBench | 0.6531 | 0.5197 | 0.4461 | 0.4286 | 0.4021 |
| | MMBench | 0.8574 | 0.7846 | 0.7238 | 0.7165 | 0.6985 |
| | MME | 0.9572 | 0.8540 | 0.8338 | 0.7434 | 0.5528 |
| **CC3M (3M)** | SEEDBench | 0.3540 | 0.2622 | 0.2060 | 0.2165 | 0.1912 |
| | MMBench | 0.6871 | 0.5327 | 0.4318 | 0.4018 | 0.4191 |
| | MME | 0.6213 | 0.4984 | 0.5001 | 0.4314 | 0.3909 |
| **COCO-Caption (0.1M)** | SEEDBench | 0.9501 | 0.6952 | 0.6429 | 0.6354 | 0.5994 |
| | MMBench | 0.8386 | 0.7410 | 0.6347 | 0.7143 | 0.7085 |
| | MME | 0.8121 | 0.7227 | 0.6533 | 0.6892 | 0.7121 |

Table 14: The text contamination rate about the original, and L1, L2, L3, L4 strategy between benchmarks and pre-training dataset.

---

Wolfe, Jeremy M. "Visual attention." Seeing (2000): 335-386.

Xu, Jiang, et al. "Language in context: emergent features of word, sentence, and narrative comprehension." *Neuroimage* 25.3 (2005): 1002-1015.

We further detect the text contamination rates of the newly generated questions after strategy l1, l2,l3,l4, based on the MME, SEEDBench, and MMBench benchmarks. Together with the original text contamination rate among contaminated images for comparison, the experimental results are shown in the table 14. As can be seen, the text contamination rates of our l1, l2,l3,l4 question variants have decreased on three pretraining sets LAION(100M), CC3M(3M), and COCO-Caption (0.1M). These prove that our newly generated questions are not found in the pretraining data, successfully relieved the data contamination issue in existing static benchmarks.

## A.13 COMPATIBILITY OF VLB ON OTHER LVLM TASKS

Since our method can dynamically transform both images and text to generate new samples, we believe our framework can also be applied to other tasks that VLMs can perform. Below, we conduct experiments on a common multi-modal task of LVLM: image caption. Firstly, we select the famous image caption benchmark, COCO Caption, and randomly sample a 500 pairs subset. Secondly, we use our three visual strategies to generate new images on COCO Caption: adding new objects($\mathcal{V}_1$), removing existing objects($\mathcal{V}_2$), and outpainting($\mathcal{V}_3$). Next, for each newly generated image, we use GPT-4V to generate corresponding captions, composing new <image, caption > pairs. Finally, we evaluate the original and the three dynamic COCO-caption variants via Evalkit on 6 popular VLMs.

| Model | Vanilla | $\mathcal{V}_1$ | $\mathcal{V}_2$ | $\mathcal{V}_3$ |
|---|---|---|---|---|
| TransCore-M | 65.4 | 60.1 | 62.4 | 59.1 |
| DeepSeek | 58.6 | 46.1 | 51.0 | 45.2 |
| InternVL-2 | 63.8 | 59.7 | 65.9 | 60.3 |
| XComposer2 | 70.1 | 65.4 | 68.6 | 67.6 |
| Monkey-Chat | 68.5 | 61.6 | 65.9 | 62.8 |
| Qwen-VL-Chat | 39.8 | 40.8 | 42.5 | 36.3 |

Table 15: The result of image caption task on the original, and $\mathcal{V}_1$, $\mathcal{V}_2$, $\mathcal{V}_3$ strategy in COCO-Caption.

As can be seen from the table 15, our newly generated samples achieve similar results to the original dataset. And similar conclusions can be drawn, namely that strategy ($\mathcal{V}_2$) is simpler than ($\mathcal{V}_1$) and ($\mathcal{V}_3$). This demonstrates that our dynamic framework can also effectively be applied to other multi-modal tasks, allowing a more flexible and comprehensive evaluation of LVLMs.

## A.14 FAIL CASES BEFORE JUDGE MODULE

In our human verification process, it is observed that the V1 strategy leads to a much higher fail-to-detect rate before passing the judge module. We thought this is primarily because, in a small portion of generated images, the addition of objects causes partial occlusion of the core objects related to the question. We selected two fail cases from MMBench $\mathcal{V}_1$ strategy, with the types of being attribute comparison and celebrity recognition, respectively. As can be seen in Figure 15 (a), the added laptop covers most of the cat, thus affecting the comparison of animal sizes. In Figure 15 (b), the added balloon slightly obscures the person's face, which might impact the recognition of the person. Therefore, images with partial occlusions like those are filtered out by the judge module, which reflects the rigor and effect of the judge module.

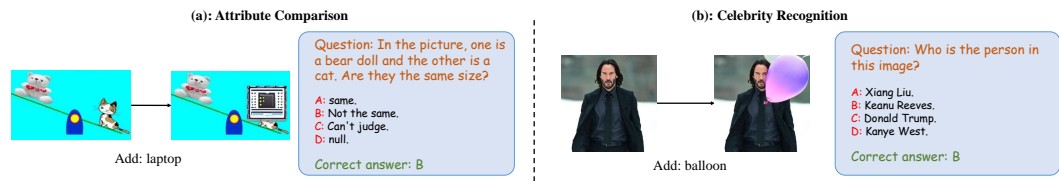

Figure 15: (a) The added laptop covers most of the cat, thus affecting the comparison of animal sizes. (b) The added balloon slightly obscures the person's face, which might impact the recognition of the person.

Chen, Xinlei, et al. "Microsoft coco captions: Data collection and evaluation server." *arXiv preprint arXiv:1504.00325* (2015).

