# OpenReview forum: "Dynamic Multimodal Evaluation with Flexible Complexity by Vision-Language Bootstrapping"
_ICLR.cc/2025/Conference — ICLR 2025 Oral_

### Official Review · Reviewer_jqNx · 2024-10-21

**Soundness:** 3
**Presentation:** 1
**Contribution:** 3
**Rating:** 6
**Confidence:** 4

**Summary:**

The paper introduces a dynamic multimodal evaluation strategy to avoid data contamination for benchmarks. It shows that current popular evaluation benchmarks exist image-only contamination and image-text contamination. Image bootstrapping and language bootstrapping method can be used separately or in combination to adjust the difficulty of the questions while keeping the answers unchanged. Several experiments on 3 benchmarks and 8 VLMs are provided to verify the author's point of view.

**Strengths:**

1. The paper proposes a novel dynamic multimodal evaluation framework VLB, which has a a flexible complexity adjustment evaluation mechanism.
2. By editing images and modifying questions, a set of evaluation samples with different levels of complexity can be generated. This method can be used to probe the upper and lower bounds of VLMs' capabilities in certain tasks.
3. The VLB framework is highly versatile. It can be easily plugged in a lot of benchmarks.

**Weaknesses:**

1. Authors use many models, such as GPT-4V, PowerPaint, SAM, to insert and remove objects from original images and employ GPT-4 for language reconstruction. Each step requires consistency evaluation, which may increase the complexity of implementation and lead to more unexpected errors or inconsistencies.
2. Generating new visual and language samples takes a long time, especially during large-scale evaluations. It also requires significant computational resources to produce new samples.
3. The VLB framework relies on PowerPaint, SAM and GPT-4V to generate new visual and language samples.
4. Although DME can generate diverse samples, the performance of VLMs may decrease on these samples, and the specific reasons for this are not clear.
5. Lack visualization for error analyses.

**Questions:**

1. In practice, how can you ensure that when generating a lot of new samples, changes in images and questions do not introduce unnecessary bias or semantic errors?
2. Are there any automated validation methods that can guarantee semantic consistency across all generated samples?
3. When the external tools, such as PowerPaint, SAM and GPT-4V, being relied upon have limitations, for example, sometimes there will be some artifacts after using PowerPaint to remove an object, how does the DME ensure the fairness and accuracy of the evaluation?
4. How to improve the interpretability of evaluation results?
5. How to assist researchers in better understanding the specific areas where the model underperforms?
6. Can you provide the specific VLMs mentioned in Table 1? For example, InternVL2-1B or InternVL2-2B or … InternVL2-76B.
7. In Figure 4 “image outpainting” part, the proportion of the floor in the image V3 is larger than in the Vanilla image, and the question is “What type of flooring does the kitchen have?”, so in this situation, can “image outpainting” be defined as a hard task?
8. Have you resized the image when using PowerPaint and GPT-4V?

---

> ### Author Response · Authors · 2024-11-21
> **Response Part 1**
>
> Dear Reviewer jqNx,
>
> Thanks for your thorough review and comments on our work. Below, we will clarify your doubts and address your concerns on each point.
>
>
>
> ### **W1: Concerns of the complexity increase of implementation and  more unexpected errors or inconsistencies caused by each step, such as GPT-4V, PowerPaint, SAM.**
>
> Thanks for your comment, and this point is also one of our key focuses. Therefore, we have taken methodological measures to maintain consistency, and we have conducted extensive experiments to demonstrate the high credibility of generated data. On the one hand, we have indeed taken three mechanisms to ensure the consistency of the generated data, but these measures are not as complex as you imagined. We hope the illustration in **W2** would address your concern for the complexity of our framework. On the other hand, as can be seen from our experimental result, our method contains few unexpected errors.
>
> Now we will explain for you about the three mechanisms we take to maintain consistency and avoid unexpected errors:
>
> **(1)GPT instruction:** For strategies v1 and v2, when utilizing GPT4v for generating bounding-box and mask index, we have made strict and serious prompts for it, as show in **Appendix A1**. In specific, we input the original image, question, and answer into GPT and emphasize that added the object and removed object must not change the answer-related visual content. The detailed instructions, input content , and output format are also detailed in Table 4. For strategies L3 and L4, when generating context with GPT, we also input the original image, question, answer, and strict instructions. The instructions emphasize GPT to generate relevant captions or irrelevant context, and forbid GPT directly revealing the answer.
>
> **(2) Judge module:** To further ensure consistency between the original sample and the dynamic variant, we incorporate a judge module as an automated verification technique. As shown in the **Section 4.4**, the judge module can adversarially select images and questions that meet our modification requirements. Here we designed different judge prompts to check each bootstrapping strategy, detailed prompts are in **Appendix A2**. Similar to MPA[1], the judge operates in an adversarial manner and returns a  'Yes' or 'No' verdict. If the response is `No', the sample will be regenerated until it passes the judgement. If the new sample does not pass after five attempts, the original sample is used instead. We also provide you a detailed ablation study of judge module in **Q2**.
>
> **(3) Human verification**: What’s more, we further conduct a human verification to validate the effectiveness of our judge module, and to ensure that our variants maintain consistent with the original benchmark both visually and linguistically. As shown in **Appendix A5**, we recruit 20 human experts (with bachelor or higher degree) and randomly selected 100 samples from each strategy for each benchmark, totally 2100 samples. The result in Table 11 showcases a high level of alignment between the judge model and human verification, proving the equivalence and correctness of our dynamic methodology.

---

> ### Author Response · Authors · 2024-11-21
> **Response Part 2**
>
> ### **W2: Concerns of significant time and computational resources cost when producing new samples.**
>
> Our detailed experimental content may have led to your misunderstanding about the time and resources cost of our framework. In reality, our framework is time-friendly and resource-efficient.
>
> **(1) About the time consumption:**  Actually, it is very quick and convenient to generate a new sample via our framework, both visually and linguistically. We hope the specific average time consumption would address your concern. In specific, for the visual modality, as shown in **Figure 4**, creating a new image mainly requires the GPT and image editing modules. The average time to call the GPT function is 4.3937 seconds, and the average time to generate an image by PowerPaint is 2.4267 seconds. For the language modality, generating a new sample only requires one call to GPT or one rephrasing method, with the average time being 5.1965 seconds. Therefore, our proposed framework does not take a long time, instead, it is very quick and convenient.
>
> **(2) About large-scale evaluations:** As mentioned in the **Section 5.1**, we utilize the integrated platform EvalKit for all evaluation, which is a highly convenient and standardized evaluation tool. It integrates  various types of benchmarks and mainstream LVLMs. Therefore, once we have prepared the dataset, it only takes a single command line to run the evaluation by EvalKit, requiring little time and resources. Our extensive experiments may lead to your misunderstanding of heavy evaluation, but the entire evaluation process is not cumbersome but rather systematic and well-organized.
>
> **(3) About computational resources:** Our VLB mainly involves leveraging three tools (i.e., GPT, Powerpaint and SAM. Among these, GPT does not require GPU resources. SAM and PowerPaint can be run on low-specification GPUs. With these powerful tools, our VLB requires low resources to generate high quality new samples. What's more, as mentioned in (1), generating a new sample requires only few seconds.
>
>
>
> ### **W3: The VLB relies on PowerPaint, SAM and GPT-4V to generate new samples.**
>
> **(1) The reason why we chose these models.** The utilized tool models like PowerPaint, SAM, and GPT-4V are all currently the undisputed state-of-the-art in their respective fields. Therefore, to ensure the accuracy and consistency of newly generated datas, we integrate these sota and effective models into our unified framework VLB.
>
> **(2) These models are replaceable.** Any model with similar functions can be substituted. Moreover, as models of this type continue to evolve, our VLB framework can also evolve to become better.  and our framework is compatible with models of similar functionality.
>
> **(3) To show the compatibility of our framework, we conduct an ablation study, taking image editing model as an example.** We selected models embodied similar functionality with PowerPaint, namely Brushnet[3] and Stable Diffusion-outpainting[4], to implement the v1 and v3 strategies for generating image samples. The evaluation results of  different image editing models are shown in the table below.
>
> | Model       | vanilla | (v1) +powerpaint | (v1) +brushnet | vanilla | (v3) +powerpaint | (v3) +stable outpainting |
> | ----------- | ------- | ---------------- | -------------- | ------- | ---------------- | ------------------------ |
> | TransCore-M | 73.58   | 69.10 (4.48↓)    | 70.91 (2.67↓)  | 73.58   | 69.52 (4.06↓)    | 70.45 (3.13↓)            |
> | DeepSeek    | 69.44   | 64.79 (4.65↓)    | 66.26 (3.18↓)  | 69.44   | 69.39 (0.05↓)    | 69.14 (0.30↓)            |
> | InternVL-2  | 76.80   | 70.46 (6.34↓)    | 71.36 (5.44↓)  | 76.80   | 74.45 (2.34↓)    | 75.05 (1.75↓)            |
>
> Table A. The evaluation result of our VLB framework utilizing different image editing models on SEEDBench.
>
> As can be seen from the table, although there are minor numerical differences between the results from PowerPaint and other editing models, their trends are similar. Therefore, our framework is verified to be adaptable to many tool models with similar functions.

---

> > ### Author Response · Authors · 2024-11-21
> > **Response Part 3**
> >
> > ### **W4: The specific reasons for VLM performance decrease on dynamic samples.**
> >
> > Thanks for you comment. Actually, the decrease in LVLM performance is a common phenomenon across the whole LVLM research community. We hope the following clarification would address your questions.
> >
> > **(1) Firstly, it is ackonwledged by LVLM community that there is still much improvement space in the LVLM capability and generalizability across many areas**, so it is common for LVLMs to underperform in some cases. Previous works has demonstrated that many factors can lead to a considerable decrease in LVLM performance, such as challenging benchmarks[5,6,7], modifications of images[7,8], and role-play/long context question[9,10,11]. Therefore, it is quite reasonable for our newly generated samples to be chanllenging and lead to a few performance decrease since our DME involves both novel modifications of images and different questioning modes.
> >
> > **(2) From the motivation of our research, it is quite reasonable for the model's performance to vary or even decline when used by different users.** As stated in **Section 4.2** and **Section 4.3**, our visual and linguistic strategies are derived from practical situations in real user interactions, simulating how different users utilize the LVLM model. Our visual strategies mimic different levels of user visual attention. For instance, children’s attention might be more easily distracted by irrelevant objects. Similarly, users with various identities and backgrounds tend to prefer different linguistic expressions, hence they may provide inputs at the word, sentence, or context level differently. Our linguistic strategies simulate these different questioning expressions. Therefore, the user interaction scenarios we simulate is likely to be challenging for LVLMs, which is consistent with some previous works [11,12].
> >
> > **(3) Based on the visualizations presented in Appendix A6,** it is reasonable to impact LVLM's assessment. We will now provide additional visualization content in **W5** for your review.
> >
> > **(4) We have conducted further analysis on the experimental results for you in Q4 and Q5**.
> >
> >
> >
> >
> >
> > ### **W5: Visualization for error analyses.**
> >
> > In fact, we have already displayed the visualization for error analyses that you mentioned in **Appendix A6**.  Now, we will provide you a more detalied error analyses on both visual and linguistic modality.
> >
> > **(1) Visually,** as shown in **Figure 13**, we visualize the GPT4o's responses to images before and after dynamic changes. Part (a) shows an example of instance identity task, where adding a sofa caused the LVLM to wrongly answer a question it correctly responded in the original. This may be due to the added red sofa, which is brightly colored and distracts the model's attention. Part (b) is an example of spatial relation task, where the removal of a cabinet made the relative positions of the fan and stage clearer, allowing the LVLM to correctly answer the question it had previously gotten wrong.
> >
> > **(2) Linguistically,** We further visualize more of LVLMs' responses in **Figure 14**, especially about questions before and after dynamic changes. As can be seen, our dynamic strategy for generating new multi-modal VQA pairs can results in performance changes, presenting a greater challenge for LVLMs.
> >
> >
> >
> >
> >
> >
> >
> > ### **Q1: How VLB ensure that changes in images and questions do not introduce unnecessary bias or semantic errors？**
> >
> > Tanks for your comment, we also believe this assurance is important. Therefore, we have taken methodological measures to maintain consistency, and we have conducted extensive experiments to demonstrate the high credibility of generated data, which contains few unexpected bias and errors. As explained for you detailedly in **W1**.

---

> > > ### Author Response · Authors · 2024-11-21
> > > **Response Part 4**
> > >
> > > ### **Q2: Any other automated validation methods that can guarantee semantic consistency.**
> > >
> > > **(1) We have designed an adversarial judge module detailed in Section 4.4 and Appendix A2.**  The judge module is just an automated validation method  you mentioned to guarantee semantic consistency. As shown in the **Section 4.4**, the judge module can adversarially select images and questions that meet our modification requirements. Specifically, The judge module is informed what modification is conducted, and is asked to check whether the answer is still correct for the newly generated image or question. Here we designed different judge prompts to check each bootstrapping strategy, detailed prompts are in **Appendix A2**. Similar to MPA[12], the judge operates in an adversarial manner and returns a  'Yes' or 'No' verdict. If the response is `No', the sample will be regenerated until it passes the judgement. If the new sample does not pass after five attempts, the original sample is used instead.  In practice, we use InternVL-2, an open-source powerful LVLM as a capable and affordable judge.
> > >
> > > **(2) We added an ablation study of the judge module, and the results validated its effectiveness.** In specific,  we sampled data from generated images and questions that do not undergo adversarial selection by the judge module, and conducte a same human verification.
> > >
> > > |              | Dataset   | V1   | V2   | V3   | L1   | L2   | L3   | L4   |
> > > | ------------ | --------- | ---- | ---- | ---- | ---- | ---- | ---- | ---- |
> > > | before judge | SEEDBench | 83   | 92   | 91   | 89   | 93   | 93   | 98   |
> > > |              | MMBench   | 81   | 94   | 88   | 91   | 90   | 94   | 95   |
> > > |              | MME       | 84   | 95   | 87   | 87   | 85   | 95   | 97   |
> > > | after judge  | SEEDBench | 97   | 100  | 98   | 93   | 100  | 98   | 100  |
> > > |              | MMBench   | 98   | 98   | 97   | 97   | 98   | 100  | 98   |
> > > |              | MME       | 98   | 100  | 98   | 98   | 98   | 100  | 100  |
> > >
> > > Table B. The human verification result of generated samples before and after the judge module.
> > >
> > > As shown in the table above, it is evident that for most strategies, the verification rate is significantly improved after the judge module, demonstrating that our judge module is an effective automated verification technique. The increased approval rate for the visual strategy demonstrates the effective judgment of our Judge module in image-text alignment, while the improved approval rate for the linguistic strategy reflects the effective verification capability of the Judge module in terms of semantics.
> > >
> > > | Dataset   | V1     | V2     | V3     | L1     | L2     | L3     | L4     |
> > > | --------- | ------ | ------ | ------ | ------ | ------ | ------ | ------ |
> > > | SEEDBench | 0.3477 | 0.1014 | 0.2441 | 0.3428 | 0.0307 | 0.0303 | 0.0054 |
> > > | MMBench   | 0.4395 | 0.0906 | 0.2729 | 0.3092 | 0.0158 | 0.0867 | 0.0567 |
> > > | MME       | 0.4019 | 0.0973 | 0.2063 | 0.2085 | 0.3263 | 0.0548 | 0.1863 |
> > >
> > > Table C. The average adversarial iterations by our judge module for each strategy.
> > >
> > > We have also provided the average number of adversarial iterations by our judge module for each strategy. The iteration number means the average times of samples be rejected and re-selected(i.e., do not pass) by the judge model. This demonstrates that our judge module is an effective and intelligent filter, which can select  accurately modified samples that are consistent with the original, avoiding unexpected errors.

---

> ### Author Response · Authors · 2024-11-21
> **Response Part 5**
>
> ### **Q3:  sometimes there will be some artifacts after using PowerPaint to remove an object, how does the DME ensure the fairness and accuracy of the evaluation.**
>
> **(1) First, we want to explain for you the reason why we chose these models.** The utilized tool models like PowerPaint, SAM, and GPT-4V are all currently the undisputed state-of-the-art in their respective fields. Therefore, to ensure the accuracy and consistency of newly generated datas, we integrate these sota and effective models into our unified framework VLB.
>
> **(2) About some artifacts after using PowerPaint**: On one hand, Currently, image editing models have become quite advanced and are widely used. PowerPaint inherently produces fewer artifacts and is among the one of the best-performing models. The original paper of PowerPaint [13] has already demonstrated its reliability in the editing process through comparisons of CLIPScore and Local-FID for its object addition and removal capabilities. On the other hand, during our use of PowerPaint, we continued to uphold its standard for quality control. For strategy v1 and v2, we also have adopted the CLIPScore and Local-FID to set a quality threshold of 25.00 and 13.50 to evaluate the alignment of generated visual content with object name.  For strategy v3, we have adopted aesthetic score for selecting outpainting images with pleasing scenery extending content.
>
> **(3) About the fairness and accuracy of our evaluation:** In our practical experiment, for each image and language strategy, we set up five random seeds, generating five data variants. Therefore, all experimental results are the average of these five variants. Due to page limitations, we did not display the results of all five variants in paper. Below we provide the mean and standard deviation for each strategy across the five variants for you.
>
> | Dataset   | Model       | V1            | V2            | V3            | L1            | L2            | L3            | L4            |
> | --------- | ----------- | ------------- | ------------- | ------------- | ------------- | ------------- | ------------- | ------------- |
> | SEEDBench | TransCore-M | 69.10(0.5737) | 73.86(0.5026) | 69.52(1.3760) | 72.94(0.0353) | 71.67(0.0330) | 71.95(0.0599) | 72.19(0.0319) |
> |           | DeepSeek    | 64.79(0.7716) | 70.35(0.1560) | 69.39(0.1825) | 68.67(0.0523) | 68.51(0.0475) | 71.17(0.0655) | 70.16(0.0447) |
> |           | InternVL2   | 70.46(0.6834) | 77.67(0.6368) | 74.45(0.5221) | 75.26(0.0486) | 74.77(0.0487) | 77.24(0.0630) | 74.31(0.0444) |
> | MMBench   | TransCore-M | 75.37(0.6665) | 79.72(0.3891) | 76.89(0.5022) | 78.53(0.0056) | 78.62(0.0032) | 84.89(0.0050) | 78.55(0.0016) |
> |           | DeepSeek    | 75.90(0.2780) | 80.03(0.2573) | 77.96(0.2327) | 78.15(0.0037) | 78.92(0.0025) | 84.26(0.0042) | 79.07(0.0044) |
> |           | InternVL2   | 80.33(1.0671) | 89.28(0.7612) | 83.01(0.2827) | 85.38(0.0032) | 87.52(0.0015) | 89.59(0.0038) | 85.53(0.0049) |
> | MME       | TransCore-M | 83.13(1.1170) | 88.87(0.1922) | 86.15(0.6643) | 85.57(0.0298) | 86.31(0.0522) | 87.42(0.0534) | 72.92(0.0689) |
> |           | DeepSeek    | 78.16(1.2455) | 86.59(0.0841) | 82.69(0.6797) | 85.67(0.0425) | 84.05(0.0181) | 92.23(0.0478) | 71.18(0.0603) |
> |           | InternVL2   | 72.70(0.4262) | 82.31(0.7943) | 76.54(0.9789) | 77.15(0.0277) | 79.34(0.0474) | 79.52(0.0563) | 69.98(0.0455) |
>
> Table D.Mean and standard deviation (in parentheses) of accuracy across different strategies and datasets on LVLMs.
>
> As can be seen from the table, the standard deviation in each strategy is slight, thus the variance scale caused by the randomness of GPT-4V and PowerPaint is also minimal. These demonstrate the reliability of the average accuracy result reported in our paper.
>
> **(4) As a dynamic evaluation framework, our primary focus is on ensuring the consistency with the original.** Therefore, the core issue we address is whether the newly generated samples produce reliable content while maintaining the original answers unchanged. The specific pixel quality of the images generated is not our main concern. Moreover, as shown in **W3**, we have taken measures to ensure image quality and have selected high-quality images for our final evaluation dataset.

---

> > ### Author Response · Authors · 2024-11-21
> > **Response Part 6**
> >
> > ### **Q4: How to improve the interpretability of evaluation results?**
> >
> > Tanks for your comment. In W4, we have already explained some instances where the accuracy of LVLMs might indeed decrease. Wha's more, we provide two further analysis for you to prove the interpretability and rationality of our evaluation results. We value your comments and have included the two analysis into our revised paper.
> >
> > **(1) Visually, we analyzed the relationship between the clipscore difference value and the accuracy difference value**. For strategy v1, v2 and v3, we calculated the CLIPScore between newly generated images  and the original images, together with the average accuracy difference of each strategy. Below we take MMbench as an example for analysis,  and the table is ordered by the accuracy difference value compared to the original results for each strategy.
> >
> > | strategy            | V1     | V3     | V2     |
> > | ------------------- | ------ | ------ | ------ |
> > | Accuracy difference | 5.7655 | 2.2777 | 0.4133 |
> > | CLIPScore           | 0.8150 | 0.8776 | 0.9502 |
> >
> > Table E. Relationship of accuracy difference and CLIPScore on MMBench.
> >
> > As shown in the table, it can be generally found that, images with a lower CLIPScore with the original image cause greater accuracy changes for LVLMs. This indicates that the relatively bigger modifications on the original image can lead to more significant visual attention disruptions and thus pose bigger challenges to the LVLMs.
> >
> >
> >
> > **(2) Linguistically,** For strategy L1, L2, L3, L4, we calculated the word number of questions, namely question length. Then we analyzed the relationship between the question length and the accuracy differences on MMBench, as depicted in the following table.
> >
> > | strategy            | L4      | L3      | L1     | L2     |
> > | ------------------- | ------- | ------- | ------ | ------ |
> > | Accuracy difference | 3.2211  | 2.900   | 1.5944 | 1.1433 |
> > | Question length     | 85.7337 | 63.6534 | 8.8523 | 9.3083 |
> >
> > Table F. Relationship of accuracy difference and Question Length on MMBench.
> >
> > Generally, it can be observed that the longer question can lead to more accuracy changed for LVLMs. This also aligns with the conclusions in previous work[9], which reveals longer contents pose greater challenges to LVLMS. Therefore, the result of our linguistic strategy on LVLM is reasonable and systematic.
> >
> >
> >
> >
> >
> > ### **Q5: How to better understanding the specific areas where the model underperforms?**
> >
> > Thanks for your comment. As you mentioned, we explored the performance of LVLMs on different tasks with newly generated data, as shown in Figure 9. We observed that the model underperformed in some tasks, such as ‘Instance Interaction’, ‘Text Understanding’ and ‘Spatial Relation’. We have hypothesized in the paper that this may because the images in these tasks inherently contain fewer objects, and thus adding/removing an object results in greater disruption for them.
> >
> > To address your questions, we quantified the number of image objects in these tasks and compared it to the extent of underperformance by the model. Here, we use the number of masks segmented from the images to simulate the number of objects. As shown in Table below, we verified that the fewer the objects in the original image, the more the model's accuracy changed after adding/removing an object. We believe that our additional experiments can assist you in better understanding this phenomenon.
> >
> > | Category             | Accuracy Difference | Object Number |
> > | -------------------- | ------------------- | ------------- |
> > | Text Understanding   | 12.1313             | 8.3333        |
> > | Spatial Relation     | 10.7080             | 6.5303        |
> > | Instances Counting   | 9.9091              | 9.6543        |
> > | Instance Interaction | 9.9091              | 8.5000        |
> > | Instance Identity    | 7.8063              | 9.0714        |
> > | Instance Location    | 5.8646              | 10.2474       |
> > | Instance Attributes  | 5.5885              | 11.0240       |
> > | Scene Understanding  | 5.3579              | 10.3333       |
> > | Visual Reasoning     | 2.6738              | 11.6471       |
> >
> > Table G. Relationship of accuracy difference and object number in different categories on SEEDBench.

---

> > > ### Author Response · Authors · 2024-11-21
> > > **Response Part 7**
> > >
> > > ### **Q6: The specific VLMs param version in Table 1.**
> > >
> > > Of course, below is the table from Table 1 that lists the specific model names and their parameters used in our experiment.
> > >
> > > |  Model   | DeepSeek-VL | TransCore-M | Monkey-Chat | LLaVA-NeXT-Vicuna | Qwen-VL-Chat | XComposer2 | Yi-VL | InternVL |
> > > | :------: | :---------: | :---------: | :---------: | :---------------: | :----------: | :--------: | :---: | :------: |
> > > | **Size** |     7B      |     28B     |    9.8B     |        13B        |     20B      |     7B     |  34B  |    8B    |
> > >
> > > Table H. The specific model names and their parameters used in our experiment.
> > >
> > >
> > >
> > > ### **Q7:  Can ‘image outpainting’ be defined as a hard task？**
> > >
> > > Thanks for your careful reading. The reason why we regard the outpainting task as a hard task, is not decided by any single instance, but is a conclusion from overall effects, our experimental results, and our simulating motivations. Below is a detailed explanation provided for you.
> > >
> > > **(1) First,** regarding your comment that 'the proportion of the floor in the image V3 is larger than in the Vanilla image,'  In fact, since outpainting extends the entire image, although the floor area increases, the overall size of the whole image also increase in both length and width, **thus the proportion of the floor in the image V3 will not enlarge too much.**
> > >
> > > **(2) The image you referred to is just an single example.** Most VQA images have their core objects related to the answer either centered or distributed across various positions in the image. In the VQA dataset, each image scenario are uniformly distributed and cover various diversity. Therefore, a single example cannot lead to a general conclusion.
> > >
> > > **(3) Our defination of 'hard' is considered from our experimental results and motivation.** On the one hand, the defination 'hard' is derived from a series of comprehensive experimental results. According to **Table 1**, Almost all LVLMs experienced a decrease in accuracy with the image outpainting strategy, thus it can be defined as a hard task. On the other hand, image outpainting is akin to viewing a scene from varying distances; the farther away, the more likely it is that the clarity of the scene's content decreases, which in turn lowers the accuracy of related questions.
> > > ### **Q8: Have you resized the image when using PowerPaint and GPT-4V？**
> > >
> > > No, we did not perform any resizing operations on the images to ensure accurate modifications.
> > >
> > > **(1) We keep the original size when utilizing PowerPaint to edit the images.** We noticed that code in PowerPaint can choose whether to resize the image, so commented out the resizing code during our experiments to ensure that the size of the generated images remained exactly the same as the original.
> > >
> > > **(2) The images inputted into GPT-4V also retain their original sizes**, and we specifically inform GPT-4V of the exact width and height of the input images. This is detailed in **Table 4** with specific instructions.

---

> > > > ### Author Response · Authors · 2024-11-21
> > > > **Response Part 8**
> > > >
> > > > ### **References**
> > > >
> > > > [1] Zhu, Kaijie, et al. "Dynamic Evaluation of Large Language Models by Meta Probing Agents." *Forty-first International Conference on Machine Learning*. 2024.
> > > >
> > > > [2]  [open-compass/VLMEvalKit: Open-source evaluation toolkit of large vision-language models (LVLMs), support 160+ VLMs, 50+ benchmarks (github.com)](https://github.com/open-compass/VLMEvalKit)
> > > >
> > > > [3] Ju, Xuan, et al. "Brushnet: A plug-and-play image inpainting model with decomposed dual-branch diffusion." *arXiv preprint arXiv:2403.06976* (2024).
> > > >
> > > > [4] Rombach, Robin, et al. "High-resolution image synthesis with latent diffusion models." *Proceedings of the IEEE/CVF conference on computer vision and pattern recognition*. 2022.
> > > >
> > > > [5] Li, Baiqi, et al. "NaturalBench: Evaluating Vision-Language Models on Natural Adversarial Samples." *arXiv preprint arXiv:2410.14669* (2024).
> > > >
> > > > [6] Liu, Shuo, et al. "Convbench: A multi-turn conversation evaluation benchmark with hierarchical capability for large vision-language models." *arXiv preprint arXiv:2403.20194* (2024).
> > > >
> > > > [7] Zhang, Hao, et al. "Avibench: Towards evaluating the robustness of large vision-language model on adversarial visual-instructions." *arXiv preprint arXiv:2403.09346* (2024).
> > > >
> > > > [8] Wu, Xiyang, et al. "AUTOHALLUSION: Automatic Generation of Hallucination Benchmarks for Vision-Language Models." *arXiv preprint arXiv:2406.10900* (2024).
> > > >
> > > > [9] Wang, Weiyun, et al. "Needle In A Multimodal Haystack." *arXiv preprint arXiv:2406.07230* (2024).
> > > >
> > > > [10] Ma, Yubo, et al. "Mmlongbench-doc: Benchmarking long-context document understanding with visualizations." *arXiv preprint arXiv:2407.01523* (2024).
> > > >
> > > > [11] Wang, Zekun Moore, et al. "Rolellm: Benchmarking, eliciting, and enhancing role-playing abilities of large language models." *arXiv preprint arXiv:2310.00746* (2023).
> > > >
> > > > [12] Zhu, Kaijie, et al. "Dynamic Evaluation of Large Language Models by Meta Probing Agents." *Forty-first International Conference on Machine Learning*. 2024.
> > > >
> > > > [13] Zhuang, Junhao, et al. "A task is worth one word: Learning with task prompts for high-quality versatile image inpainting." *arXiv preprint arXiv:2312.03594* (2023).

---

> ### Author Response · Authors · 2024-11-25
> **Sincerely looking forward to more discussion with you**
>
> Dear Reviewer jqNx,
>
> Thank you for the precious review time and insightful comments on our paper. We have provided corresponding responses and results in great detail. If you have any other concerns, we are more than happy to provide additional clarification as well as experiments at any time. Sincerely looking forward to your reply!
>
> Best,
>
> Authors

---

> > ### Author Response · Authors · 2024-11-27
> > **Sincerely looking forward to your reply**
> >
> > Dear Reviewer jqNx,
> >
> > We sincerely appreciate your time and effort in reviewing our work. We thank your deep comprehension and suggestions, which may have strengthened our manuscript.
> >
> > To address the concerns you raised, we provide (1) a detailed summary of the three mechanisms we take to maintain consistency and avoid unexpected errors, (2) a clarification and clear numbers about the time and resource cost of our framework, which is quick and friendly, (3)  an ablation study on image editing models, to show the compatibility of our framework, (4) an explanation for VLM performance decrease, (5) more visualization for error analyses, (6) an ablation study of the judge module, validating its effectiveness, (7) the mean and standard deviation for each strategy across variants,  to verify the fairness and accuracy of our evaluation, (8) interpretable analysis of evaluation results, (9) a clarification of image resizing.
> >
> > We would appreciate the opportunity to continue dialogue with you to fully meet your expectations. If you still remain some aspects of our work unclear or have other additional concerns, We are always prepared to discuss or provide more information to you. Our commitment is to address any concerns you may have and enhance the quality of our paper.
> >
> > Best regards.

---

### Official Review · Reviewer_7UPF · 2024-11-03

**Soundness:** 3
**Presentation:** 4
**Contribution:** 3
**Rating:** 8
**Confidence:** 5

**Summary:**

This work introduces VLB, a benchmark generation strategy aiming to prevent the baseline models from being evaluated on contaminated data (data leakage in essence). VLB extends current LVLM benchmarks by employing bootstrapping strategies to create altered test cases in variable difficulties in a controllable manner. The experimental results using VLB-modified benchmarks show that data leakage is evidently prominent in existing practices, and VLB may help establish fairer baselines for LVLM evaluations.

**Strengths:**

The paper is a pleasant read and is easy to follow. The paper is written in a well structured way following a clear plot line. I particularly find the parts where the bootstrapping strategies are introduced well written, which greatly helps me understand this work on an intuitive level.

**Weaknesses:**

I do have one particular concern regarding the veracity of the VLB-modified data.

VLB strategies such as V1 (editing in a new object in the image) and L4 (adding irrelevant context in to the text) modify the original test case in a controlled manner. **However, how do we verify if the original test cases have been loyally modified in the way we want?** So far, such veracity verification steps are only observed in Figure 11 using human verification on a small batch of sampled data. The authors should consider additional automated verification techniques, such as the shift in semantic scores/image-text alignment metrics before vs after applying VLB strategies.

After all, we do not want to re-evaluate LVLMs on wrongly generated test cases, which undermine the entire effort of VLB to start with.

**Questions:**

Here I also list down a few minor suggestions.

1. The naming order of the VLB strategies could be changed so that the easy ones be introduced first. For example, it would feels more natural to first introduce L3, so that readers can more easily tell that, from Table 2, L3 is the positive alternative with additional helpful cues and lead to mostly improvement performance. The same idea applies for V2 as in Table 1.
2. The term 'data contamination' is not a coined term. I sense the authors want to describe the phenomenon that pre-training data already entail the contents for the supposedly unknown test set cases. In fact, this already has had a name **data leakage** as in (Chen et al, 2024). Let's stick with the established terminology. But feel free to correct me if the authors believe the two have any nuanced difference.

Typo: Table reference missing at Line 707.

Update 11/23: Raising my assessment according to the authors' response.

After all, I find this work very intriguing although the contribution feels a bit lackluster. However, I am open for reassessment after learning more from the exchange with the authors in the rebuttal period.

Chen et al, 2024. Are We on the Right Way for Evaluating Large Vision-Language Models?

---

> ### Author Response · Authors · 2024-11-21
> **Response Part 1**
>
> Dear Reviewer 7UPF,
>
> Thank you very much for appreciating our paper as clear and well-written. Your constructive suggestions are so valuable to us. We are honored to have the opportunity to clarify the points you raised. Below is our detailed response to answer your concerns.
>
> ### **W1: Concern regarding the veracity of the VLB-modified data, and consider additional automated verification techniques.**
>
>  We are lucky to have met such a rigorous reviewer like you, this point you raised is also something we consider very important. Therefore, We will now explain from two aspects on how our work ensures if the original test cases have been loyally modified in the way we intended.
>
> **（1）In fact, we have implemented three safeguard mechanisms to ensure the consistency and modification loyalty between the origianl sample and our dynamic sample.** But sorry for that we may not summarize these three mechanisms together although we have described  them in our paper,  which might not impress you enough. So below we will exlpain the three safeguard mechanisms for you:
>
> - **GPT instruction:** For strategies v1 and v2, when utilizing GPT4v for generating bounding-box and mask index, we have made strict and serious prompts for it, as show in Appendix A1. In specific, we input the original image, question, and answer into GPT and emphasize that added the object and removed object must not change the answer-related visual content. The detailed instructions, input content , and output format are also detailed in Table 4. For strategies L3 and L4, when generating context with GPT, we also input the original image, question, answer, and strict instructions. The instructions emphasize GPT to generate relevant captions or irrelevant context, and forbid GPT directly revealing the answer.
>
> - **Judge module:** To further ensure consistency between the original sample and the dynamic variant, we incorporate a judge module as an automated verification technique. As shown in the Section 4.4, the judge module can adversarially select images and questions that meet our modification requirements. Specifically, The judge module is informed what modification is conducted, and is asked to check whether the answer is still correct for the newly generated image or question. Here we designed different judge prompts to check each bootstrapping strategy, detailed prompts are in Appendix A2. Similar to MPA[1], the judge operates in an adversarial manner and returns a  'Yes' or 'No' verdict. If the response is `No', the sample will be regenerated until it passes the judgement. If the new sample does not pass after five attempts, the original sample is used instead.  In practice, we use InternVL-2 pro, an powerful open-source LVLM as a capable and affordable judge.
>
> - **Human verification**: What’s more, we further conduct a human verification to validate the effectiveness of our judge module, and to ensure that our variants maintain consistent with the original benchmark both visually and linguistically. As shown in Appendix A5, For each dynamic strategy, we recruit 20 human experts (with bachelor or higher degree) and randomly selected 100 samples from each strategy for each benchmark, totally 2100 samples. The specific details are in Appendix A5. And  the result in Figure 11 showcases a high level of alignment between the judge model and human verification, proving the equivalence and correctness of our dynamic methodology.

---

> > ### Author Response · Authors · 2024-11-21
> > **Response Part 2**
> >
> > **(2) To further alleviate your concerns, We now present the results of human verification on consistency before and after the judge module.**  In specific,  we sampled data from generated images and questions that do not undergo adversarial selection by the judge module, and conducte a same human verification. We have also included the table below into our revised paper.
> >
> > |              | Dataset   | V1   | V2   | V3   | L1   | L2   | L3   | L4   |
> > | ------------ | --------- | ---- | ---- | ---- | ---- | ---- | ---- | ---- |
> > | before judge | SEEDBench | 83   | 92   | 91   | 89   | 93   | 93   | 98   |
> > |              | MMBench   | 81   | 94   | 88   | 91   | 90   | 94   | 95   |
> > |              | MME       | 84   | 95   | 87   | 87   | 85   | 95   | 97   |
> > | after judge  | SEEDBench | 97   | 100  | 98   | 93   | 100  | 98   | 100  |
> > |              | MMBench   | 98   | 98   | 97   | 97   | 98   | 100  | 98   |
> > |              | MME       | 98   | 100  | 98   | 98   | 98   | 100  | 100  |
> >
> > Table A. The human verification result of generated samples before and after the judge module.
> >
> > As shown in the table above, it is evident that for most strategies, the verification rate is significantly improved after the judge module, demonstrating that our judge module is an effective automated verification technique. The increased approval rate for the visual strategy demonstrates the effective judgment of our Judge module in image-text alignment, while the improved approval rate for the linguistic strategy reflects the effective verification capability of the Judge module in terms of semantics.
> >
> > | Dataset   | V1     | V2     | V3     | L1     | L2     | L3     | L4     |
> > | --------- | ------ | ------ | ------ | ------ | ------ | ------ | ------ |
> > | SEEDBench | 0.3477 | 0.1014 | 0.2441 | 0.3428 | 0.0307 | 0.0303 | 0.0054 |
> > | MMBench   | 0.4395 | 0.0906 | 0.2729 | 0.3092 | 0.0158 | 0.0867 | 0.0567 |
> > | MME       | 0.4019 | 0.0973 | 0.2063 | 0.2085 | 0.3263 | 0.0548 | 0.1863 |
> >
> > Table B. The average adversarial iterations by our judge module for each strategy.
> >
> > We have also provided the average adversarial iterations by our judge module for each strategy. The iteration number means the average times of samples be rejected and re-selected(i.e., do not pass) by the judge model. This demonstrates that our judge module is an effective and intelligent filter, which can select  accurately modified samples that are consistent with the original.

---

> > > ### Author Response · Authors · 2024-11-21
> > > **Response Part 3**
> > >
> > > ### **Q1: The naming order suggestion of the VLB strategies  that the easy ones could be introduced first.**
> > >
> > > Thanks very much for your careful reading! Your valuable comment about re-ordering truly inspired us. We will rearrange the introduction of strategies by difficulty level in the final version, because we worry that re-order now might disrupt/confuse other reviewers' and AC's understanding.
> > >
> > > We are so lucky to meet such a insightful reviewer like you! And we promise to re-order the strategies as you suggested in the final version.  We take every comment you have made very seriously and will try our best to ensure it meets the high standards expected for publication.
> > >
> > >
> > >
> > >
> > >
> > > ### **Q2: The phenomenon naming suggestion of 'data contamination' and 'data leakage'.**
> > >
> > > Thanks very much for your careful reading. Your points are very well-made, reflecting that you are undoubtedly a meticulous and knowledgeable scholar. For this reason, We specifically surveyed some related works and found that 'data leakage' and 'data contamination' sometimes have similar meanings. **As a result, we have highlighted the issue of data leakage you mentioned in the revised paper.**
> > >
> > > (1) Following your reminder,  **we have surveyed issues about data leakage and data contamination and found that both terms are widely used in reseach of LLM and LVLM**. The term 'data leakage' is used in papers such as  [2, 3], while 'data contamination' is also used in papers such as [4, 5, 6, 7]. Both the two terms are used to describe the phenomenon that pre-training data already entail the contents for the supposedly unknown test set cases.
> > >
> > > (2) We have also noted the paper[2] you mentioned, which is cited in our related work. The data leakage described in that paper is limited to textual modality of LVLMs, leaving image modality unexplored and leaving the specific contamination rate undetected. However, our work include detection of accurate data contamination rates in both images and texts modalities as Section 3. **Therefore our defined data contamination could be more comprehensive and quantifiable than the data leakage in [2]**.
> > >
> > > (3) Thank you for your preciseness. We have denoted in our paper that 'data leakage' and 'data contamination' are interchangeable sometimes in related work. And we have further emphasized the data leakage content in our revised paper. **Benefit from your suggestions, the updated version of the paper has been thoroughly verified, and we will try our best to ensure it meets your high standards.**
> > >
> > >
> > >
> > >
> > >
> > >
> > >
> > >
> > >
> > > ### **References**
> > >
> > > [1] Zhu, Kaijie, et al. "Dynamic Evaluation of Large Language Models by Meta Probing Agents." *Forty-first International Conference on Machine Learning*. 2024.
> > >
> > > [2] Chen et al, 2024. Are We on the Right Way for Evaluating Large Vision-Language Models?
> > >
> > > [3] Ghosh, Sreyan, et al. "VDGD: Mitigating LVLM Hallucinations in Cognitive Prompts by Bridging the Visual Perception Gap." *arXiv preprint arXiv:2405.15683* (2024).
> > >
> > > [4] Golchin, Shahriar, and Mihai Surdeanu. "Data contamination quiz: A tool to detect and estimate contamination in large language models." *arXiv preprint arXiv:2311.06233* (2023).
> > >
> > > [5] Li, Yucheng, et al. "An Open-Source Data Contamination Report for Large Language Models." *Findings of the Association for Computational Linguistics: EMNLP 2024*. 2024.
> > >
> > > [6] Fan, Lizhou, et al. "NPHardEval4V: A Dynamic Reasoning Benchmark of Multimodal Large Language Models." *arXiv preprint arXiv:2403.01777* (2024).
> > >
> > > [7] Zhu, Kaijie, et al. "Dyval: Graph-informed dynamic evaluation of large language models." *arXiv e-prints* (2023): arXiv-2309.

---

> > > ### Comment · Reviewer_7UPF · 2024-11-23
> > > **Nicely done.**
> > >
> > > I commend the authors' effort in addressing the reviewers' concerns. I am happy to raise my original ratings.
> > >
> > > One minor follow-up question - I notice V1, the hard strategy where we add a new object to the image, lead to a much higher fail-to-detect rate by the judge module. I wonder how such fail cases look like and if they have any pattern that can imply potential biases of the judge module. It's okay if the authors haven't had them given the time limit. But if the authors do have such concrete examples, please put them in the appendix.

---

> ### Author Response · Authors · 2024-11-24
>
> Dear Reviewer 7UPF,
>
> Thanks very much for your careful attention and response! We are lucky to have met such a rigorous reviewer like you! And we take every comment you made very seriously.
>
> As you mentioned, in our human verification process, it is observed that the V1 strategy leads to a much higher fail-to-detect rate before passing the judge module. We thought this is primarily because, in a small portion of generated images, the addition of objects causes partial occlusion of the core objects related to the question. In order to give you a more intuitive understanding, we have updated the revised paper and added visualizations of failed cases in Appendix Figure 15.  We selected two fail cases from the MMBench V1 strategy, with the types of being attribute comparison and celebrity recognition, respectively. As can be seen in figure 15 (a), the added laptop covers most of the cat, thus affecting the comparison of animal sizes. In figure 15 (b), the added balloon slightly obscures the person's face, which might impact the recognition of the person. Therefore, images with partial occlusions like those are filtered out by the judge module, which we thought primarily reflects the rigor and effect of the judge module, and may not imply potential biases. Your comments are very revealing, and we will try our best to meet your high standards.
>
> We are deeply grateful for your interest in our work and for your valuable comments, which have been instrumental in refining our paper. It is truly our luck to discuss with such a meticulous and knowledgeable reviewer as you!

---

### Official Review · Reviewer_RXK7 · 2024-11-03

**Soundness:** 3
**Presentation:** 3
**Contribution:** 3
**Rating:** 8
**Confidence:** 4

**Summary:**

This work proposes a new evaluation protocol, Vision-Language Bootstrapping (VLB), for comprehensively evaluating large vision-language models (LVLMs) with reduced data contamination and dynamic difficulty. Existing benchmarks like MM-Bench are constructed from fixed image-question pairs, which are partially observed in the training procedure of LVLMs. In VLB, both the image and the question can be modified to change the difficulty and avoid answering by memorization, while preserving the consistency with the original answer. Different operations in changing the image or question result in a series of difficulty levels. Extensive experiments are conducted to show performance change with VLB, through which the work poses new challenges for LVLMs.

**Strengths:**

1. Thorough experiment results validate the dynamic evaluation protocol VLB. First, a judge model is introduced to ensure that the dynamic image-question pair is still consistent with the original answer. Second, human examination on 2,100 samples verifies that less than 5% samples would introduce inconsistency.

2. The composition of multiple strategies effectively reduces data contamination and enables a wide range of difficulty levels. VLB can serve as a more reliable evaluation protocol than the traditional static ones.

3. The new evaluation protocol can be readily combined with existing LVLM benchmarks.

**Weaknesses:**

1. The major concern lies in the performance variance. Even with the same image-question sample and the same bootstrapping strategy, different dynamic samples can be generated, due to the randomness in GPT-4V and PowerPaint. However, the experiments do not show the scale of this variance caused by randomness. If this variance is large, the performance metrics may be less reliable.

2. Although the human verification (Figure 11) shows high consistency for each bootstrapping strategy, it is unclear if the consistency remains with composition of multiple strategies (e.g., V1+V3+L4). The errors may accumulate, and more changes to the image-question pair tend to break the original consistency.

3. The bootstrapping strategies rely on GPT-4V and PowerPaint. If they are replaced/combined with models with similar functions, can similar observations remain?

4. Some minor suggestions that do not affect the rating:
    - The bootstrapping strategies may be reordered to reflect the difficulty level. For example, how about making "removing existing objects" V1 and "adding new objects" V3?
    - In Figure 3(c), L3 should be "add relevant context."

**Questions:**

Please check the weakness section above.

---

> ### Author Response · Authors · 2024-11-21
> **Response Part 1**
>
> Dear Reviewer RXK7,
>
> Thank you for your valuable comments. We appreciate the chance to clarify and address your concerns, which we believe will enhance our paper. Below, we respond in detail to each point you raised:
>
> ### **W1: The scale of the performance variance caused by randomness in GPT-4V and PowerPaint.**
>
> Thanks for your deep thought, we also considered it important this and have taken some measures. In our practical experiment, for each image and language strategy, we set up five random seeds, generating five data variants. Therefore, all experimental results are the average of these five variants. Due to page limitations, we did not display the results of all five variants in paper. Below we provide the mean and standard deviation for each strategy across the five variants for you. We value your comment and have included the table in our revised paper.
>
> | Dataset   | Model       | V1             | V2             | V3            | L1            | L2            | L3            | L4            |
> | --------- | ----------- | -------------- | -------------- | ------------- | ------------- | ------------- | ------------- | ------------- |
> | SEEDBench | TransCore-M | 69.10 (0.5737) | 73.86 (0.5026) | 69.52(1.3760) | 72.94(0.0353) | 71.67(0.0330) | 71.95(0.0599) | 72.19(0.0319) |
> |           | DeepSeek    | 64.79 (0.7716) | 70.35(0.1560)  | 69.39(0.1825) | 68.67(0.0523) | 68.51(0.0475) | 71.17(0.0655) | 70.16(0.0447) |
> |           | InternVL2   | 70.46(0.6834)  | 77.67(0.6368)  | 74.45(0.5221) | 75.26(0.0486) | 74.77(0.0487) | 77.24(0.0630) | 74.31(0.0444) |
> | MMBench   | TransCore-M | 75.37(0.6665)  | 79.72(0.3891)  | 76.89(0.5022) | 78.53(0.0056) | 78.62(0.0032) | 84.89(0.0050) | 78.55(0.0016) |
> |           | DeepSeek    | 75.90(0.2780)  | 80.03(0.2573)  | 77.96(0.2327) | 78.15(0.0037) | 78.92(0.0025) | 84.26(0.0042) | 79.07(0.0044) |
> |           | InternVL2   | 80.33(1.0671)  | 89.28(0.7612)  | 83.01(0.2827) | 85.38(0.0032) | 87.52(0.0015) | 89.59(0.0038) | 85.53(0.0049) |
> | MME       | TransCore-M | 83.13(1.1170)  | 88.87(0.1922)  | 86.15(0.6643) | 85.57(0.0298) | 86.31(0.0522) | 87.42(0.0534) | 72.92(0.0689) |
> |           | DeepSeek    | 78.16(1.2455)  | 86.59(0.0841)  | 82.69(0.6797) | 85.67(0.0425) | 84.05(0.0181) | 92.23(0.0478) | 71.18(0.0603) |
> |           | InternVL2   | 72.70(0.4262)  | 82.31(0.7943)  | 76.54(0.9789) | 77.15(0.0277) | 79.34(0.0474) | 79.52(0.0563) | 69.98(0.0455) |
>
> Table A. Mean and standard deviation (in parentheses) of accuracy across different strategies and datasets on LVLMs.
>
> As can be seen from the table, the standard deviation in each strategy is slight, thus the variance scale caused by the randomness of GPT-4V and PowerPaint is also minimal. These demonstrate the reliability of the average accuracy result reported in our paper.
>
>
>
> ### **W2: The consistency remains with composition of multiple strategies.**
>
> Thanks you for your careful consideration. We indeed conducted human verification only for single strategy, as we thought that if both images and texts individually ensure consistency with the original one, then the combination of them should also be consistent with the original pair.
>
> However, we value your opinions, so we selected two compositions(V1+L4, V1+V3+L4) of SEEDBench to conduct a same human verification as single strategy. In specific, we randomly select drew 100 samples from each composition, together with the original 100 sample, totaling 400 VQA samples. Similarly as **Appendix A5**, we recruit 20 human experts(with bachelor or higher degree), presenting them both the multi-strategy and original samples, and asked them (1) whether these two samples maintained consistency, and (2) whether the two samples corresponded to the same correct answer. We required them to respond with only 'yes' or 'no' . Experts independently reviewed samples. Once all experts completed their evaluations, a voting process ensued. A sample was only considered to have passed if more than half of the experts deemed it consistent with the original. The results are displayed in the table below.
>
> | Dataset   | V1   | V1+L4 | V1+V3+L4 |
> | --------- | ---- | ----- | -------- |
> | SEEDBench | 96   | 92    | 85       |
>
> Table B. The human verification result of multiple strategies on SEEDBench.
>
> From the table, there is a slight decrease in consistency as the number of compositional strategy increasing. But compared to the considerable decrease in LVLM accuracy shown in figure 6, this decrease is minimal and can be somewhat disregarded(about average 36% decline). Therefore, our conclusion remains reliable that the composition of multi-strategies poses a greater challenge for LVLM.

---

> > ### Author Response · Authors · 2024-11-21
> > **Response Part 2**
> >
> > ### **W3:  If GPT-4V and PowerPaint are replaced/combined with similar function models, can similar observations remain？**
> >
> > **(1) First, We would like to explain why we chose GPT-4V and PowerPaint as our tool models.** GPT-4V, with its powerful perception and reasoning capabilities, is undoubtedly the state-of-the-art LVLM model, thus can ensure the consistency of our newly generated sample to the greatest extent. PowerPaint is also one of the most outstanding image editing models, capable of high image quality and prompt-image alignment. Moreover, it is an unified model which integrate fuctions of object addition, object removal, and image outpainting, which meets our requirment well, allowing our framework to efficiently execute three different image strategies using the same one editing model.
> >
> > **(2) Moreover, we highly value your opinions and conduct verification experiments for you.** To explore our framework on other models with similar functions, we selected other two popular image editing models to replace PowerPaint. For strategy V1, we utilize BrushNet[1] for object addition step and remain all other steps exactly the same with PowerPaint. For strategy V3, we apply Stable Diffusion[2] for image outpainting with same riato 1.5x. The table below shows the evaluation results of our framework with PowerPaint,  BrushNet, and Stable Diffusion on SEEDBench across three LVLMs. It is worth noting that we also generated five variants for each settings and take their average results.
> >
> > | Model       | vanilla | (V1) +powerpaint | (V1) +brushnet | vanilla | (V3) +powerpaint | (V3) +stable outpainting |
> > | ----------- | ------- | ---------------- | -------------- | ------- | ---------------- | ------------------------ |
> > | TransCore-M | 73.58   | 69.10 (4.48↓)    | 70.91 (2.67↓)  | 73.58   | 69.52 (4.06↓)    | 70.45 (3.13↓)            |
> > | DeepSeek    | 69.44   | 64.79 (4.65↓)    | 66.26 (3.18↓)  | 69.44   | 69.39 (0.05↓)    | 69.14 (0.30↓)            |
> > | InternVL-2  | 76.80   | 70.46 (6.34↓)    | 71.36 (5.44↓)  | 76.80   | 74.45 (2.34↓)    | 75.05 (1.75↓)            |
> >
> > Table C. The evaluation result of our VLB framework utilizing different image editing models on SEEDBench.
> >
> > As can be seen from the table, although there are minor numerical differences between the results from PowerPaint and other editing models, their trends are similar. Therefore, the experimental conclusions in our original paper are valid and reliable. What's more, since our framework is verified to be adaptable to many tool models with similar functions, then our framework will perform better and better as these tool models are improving. We value your comment and have included the above table in our revised paper.
> >
> >
> >
> > ### **W4: Minor suggestions that do not affect the rating.**
> >
> > Thanks very much for your careful reading. We are lucky to have met such a rigorous reviewer like you.
> >
> > **(1) Your valuable comment about re-ordering inspired us.** We will rearrange the introduction of strategies by difficulty level in the final version, because we worry that re-order now might disrupt/confuse other reviewers' and AC's understanding. We are so lucky to meet such a insightful reviewer like you! And we promise to re-order the strategies as you suggested in the final version.
> >
> > **(2) Thanks to your reminder.** We are sorry for the minor spelling errors in the figure 3(c), and we have taken careful steps to reinspect all the sentences in our paper.  The updated version of the paper has been thoroughly verified, and we will try our best to ensure it meets the high standards expected for publication.
> >
> >
> >
> > ### **References**
> >
> > [1] Ju, Xuan, et al. "Brushnet: A plug-and-play image inpainting model with decomposed dual-branch diffusion." *arXiv preprint arXiv:2403.06976* (2024).
> >
> > [2] Rombach, Robin, et al. "High-resolution image synthesis with latent diffusion models." *Proceedings of the IEEE/CVF conference on computer vision and pattern recognition*. 2022.

---

> > > ### Comment · Reviewer_RXK7 · 2024-11-25
> > >
> > > Thank you very much for the detailed responses and manuscript updates. I would like increase my rating.

---

> > > > ### Author Response · Authors · 2024-11-25
> > > >
> > > > Dear Reviewer RXK7,
> > > >
> > > > Thanks very much for your thorough review and valuable comments, which have been instrumental in refining our paper. It is truly our luck to meet a meticulous and knowledgeable reviewer like you!  We are deeply grateful!

---

### Official Review · Reviewer_Rays · 2024-11-05

**Soundness:** 3
**Presentation:** 3
**Contribution:** 3
**Rating:** 8
**Confidence:** 4

**Summary:**

The paper seeks to change the static nature and data contamination of benchmarks for vision-language models.  The paper introduces VLB: vision-langauge bootstrapping that dynamically generates new visual question answering samples via bootstrapping.  The goal is for the evaluation protocol to evolve with VLM capabilities.  The paper finds that existing VLMs struggle on the new benchmark.

**Strengths:**

1. The paper identifies an important research direction: existing benchmarks are static and because of large-scale pretraining data, it is hard to verify is some test data has leaked into the pretraining or training data. This makes evaluation difficult and the paper seeks to develop a new paradigm for evaluation.
2. The idea of using insights from user interactions to inform the transformations V and L is interesting.
3. The experiments are comprehensive.

**Weaknesses:**

1. The role of user interaction is not defined in detail.  See Q1.
2. Question rephrasing has been previously explored in several other works on VQA (eg. VQA Rephrasings dataset) or robustness work such as VQA-LOL, VQA-Subquestions and others. What is the overlap of the proposed work with those benchmarks?
3. The work focuses only on VQA but there are several tasks that VLMs can perform.  Can the framework also handle capabilities that have to be evaluated without VQA?

**Questions:**

1. For example in figure 3(a), how is the visual attention and linguistic understanding converted into V1, V2, L1, L2 etc.?
2. How is it verified that the generated questions are not found in the pretraining data? Does using the VLB method ensure that? This is not discussed in the analysis.

---

> ### Author Response · Authors · 2024-11-21
> **Response Part 1**
>
> Dear Reviewer Rays,
>
> Thank you for appreciating our paper as comprehensive and identifing an important research direction, your constructive comments and suggestions are valuable for us. Below is our detailed response to clarify the points and answering the concerns you raised.
>
>
>
> ### **W1&Q1: The detailed visual attention and linguistic understanding strategy converted into V1, V2, L1, L2.**
>
> Thanks for your interest in the details of our dynamic strategy. I’m honored to explain the conversion details for you. Take Figure 3 as an example, below is how we implement user interaction changes from both visual attention and linguistic understanding perspective. **Figure 3(a)** has been modified accordingly.
>
> The user interaction with LVLM is defined by the cognitive process through which users comprehend the image and ask questions of their interest in the VQA setting. This process is influenced by visual attention and linguistic understanding.
>
> Specifically, **Visual attention** affects the way how individuals selectively focus on specific visual elements within an image or scene while filtering out less relevant information. Since different users pose various identities and backgrounds, their levels of visual attention[1] also vary. For instance, children's attention might be more easily distracted by irrelevant objects in images than educated adults. We categorize visual attention with concentration and distraction. The attention concentration is implemented by object removal while distraction is implemented by object addition and image outpainting.
>
> **Linguistic understanding**  indicates how individuals interpret and comprehend questions. This process influences the language proficiency of different users. Specifically, when different users utilize LVLMs, they may have different linguistic expressions towards a same question due to their distinct identities or educational backgrounds. Therefore, as shown in Figure 3(a), our work conducts transformation on the original question from three linguistic levels[2]: word-level (L1), sentence-level (L2), and context-level (L3, L4).
>
> Here we detailedly explain the example in **Figure 3(a)** for you:
>
> **(1) Visually,** as described in **section 4.2**, since different users pose various identities and backgrounds, their levels of visual attention[1] also vary. For instance, children's attention might be more easily distracted by irrelevant objects in images than educated adults. Therefore, as shown in Figure 3(a), we simulate the distraction, simplification, and expansion on visual attention for image bootstrapping, respectively corresponding to our three strategies : adding new objects(V1), removing existing objects(V2), and expanding original images(V3).
>
> Specifically in **figure 3(c)**, compared to the original image, we added a plant, removed a wall, and outpainted the image with a ratio of 1.5, respectively, obtaining three images after visual dynamic strategies V1,V2,V3.
> **(2) LiLinguisticallyas described in **section 4.3**, we also simulate the language usage on LVLMs from different users. Specifically, when different users utilize LVLMs, they may have different linguistic expressions towards a same question due to their distinct identities or educational backgrounds. Therefore, as shown in Figure 3(a), our work conducts transfomations on the original question from three linguistic levels[2]: word-level (L1), sentence-level (L2), and context-level (L3, L4) .
>
> For example in **figure 3(c)**, the original question is 'What type of flooring does the kitchen have?'. From the word-level, some users might habitually use the word 'cookroom' instead of 'kitchen'. From the sentence level, more literary users like writers, might phrase it as 'is equipped with what kind of' instead of 'have what type of'. From the context level, more talkative users like teachers pretend to guide into the question by giving some scene description aforehead, which corresponds to our L3 strategy 'adding relevant context'. Meanwhile, children users, who may lack mature logical thinking ability, might first ramble about unrelated gadgets in the image before posing his realistic question to LVLMs. This is corresponds to L4 strategy 'adding irrelevant context'.
>
> **Generally,** our dynamic strategies effectively simulate various real user interaction scenarios from both visual and linguistic perspectives, which is very significant for the practical application and widespread adoption of LVLMs.

---

> > ### Comment · Reviewer_Rays · 2024-11-22
> > **Reviewer Rays' response for W1&Q1**
> >
> > Thanks for the detailed explanation with examples.  I strongly recommend adding it to the main paper (and not just to explain it to me, the reviewer). It will help readers understand the contributions better.

---

> ### Author Response · Authors · 2024-11-21
> **Response Part 2**
>
> ### **W2: The difference and overlap of our proposed linguistic rephrasing strategy with those other benchmarks.**
>
> Thanks for your valuable comment, we are lucky to have met such a rigorous reviewer like you. We have carefully read the works you mentioned and analyzed the overlap and differences between our rephrasing strategy and these methods one by one. Furthermore, we value your comment and have included these works in the related work of our revised paper.
>
> **(1) Compared to the VQA Rephrasings dataset[3]:**  The VQA Rephrasings dataset is collected by manual rewriting based on the static VQA v2.0 dataset. This manual rewriting process is not only time-consuming and labor-intensive, but is also unable to extend to other benchmarks automatically. **In contrast**, our rephrasing strategy is a fully automatic process, allowing us to effectively generate new rephrased questions based on various existing multi-modal benchmarks.
>
> **(2) Compared to the VQA-LOL dataset[4]:** The VQA-LOL dataset can only transform questions in the format of  yes/no, because it used logical compositions (negation, disjunction, conjunction, and antonyms) to generate question and answer.  Only in yes/no questions can they obtain the correct paired answers through logical compositions. However, yes/no questions only represent one format of multi-modal benchmarks for LVLMs. More formats like multiple-choice and VQA have been proposed for evaluating LVLMs. **In contrast**, our rephrasing strategy l2 ‘role-playing’ has broader applicability and can be used across various question types including yes/no, multiple-choice, and VQA.
>
> **(3) Compared to the VQA-Subquestions dataset[5]:** The VQA-Subquestions dataset has the above two limitations. Each sub-question in the VQA-Subquestions dataset is generated by three unique workers, and they hired a total of 463 workers to complete the annotation tasks. This process is time-consuming and labor-intensive,  and is hard be applied to other datasets. What's more, the VQA-Subquestions dataset also merely includes yes/no questions, lacking transferability to other question formats. **In contrast**, our rephrasing strategy can effectively address the above two limitations.
>
> **(4) Most importantly:** Our rephrasing strategy better reflects our perspective of user interactions. We simulate how different users with distinct identities or educational backgrounds might pose questions when using a VLM. It aligns more closely with real-world scenarios and facilitates the practical application of LVLMs.
>
> ### **W3: Can the framework also handle capabilities that have to be evaluated without VQA？**
>
> Thanks for your constructive suggestions. Since our method can dynamically transform both images and text to generate new samples, we believe our framework can also be applied to other tasks that VLMs can perform. Below, we conduct experiments on a common multi-modal task of LVLM: image caption.
>
> Firstly, we select the famous image caption benchmark, COCO Caption[6], and randomly sample a 100 pairs subset. Secondly, we use our three visual strategies to generate new images on COCO Caption: adding new objects(v1), removing existing objects(v2), and outpainting(v3). Next, for each newly generated image, we use GPT-4V to generate corresponding captions, composing new <image, caption> pairs. Finally, we evaluate the original and the three dynamic COCO Caption datasets on 3 popular VLMs. The results are shown in the table below.
>
> | Model       | Vanilla | V1   | V2   | V3   |
> | ----------- | ------- | ---- | ---- | ---- |
> | TransCore-M | 66.0    | 56.5 | 62.7 | 60.4 |
> | DeepSeek    | 59.2    | 42.3 | 54.4 | 40.9 |
> | InternVL2   | 62.4    | 57.9 | 63.7 | 58.6 |
>
> Table A. The result of image captioning task about the original, and V1, V2, V3 strategy in COCO-Caption.
>
> As can be seen from the table, our newly generated samples achieve similar results to the original dataset. And similar conclusions can be drawn, namely that strategy V2 is simpler than V1 and V3. This demonstrates that our dynamic framework can also effectively be applied to other multi-modal tasks, allowing a more flexible and comprehensive evaluation of LVLMs.

---

> > ### Author Response · Authors · 2024-11-21
> > **Response Part 3**
> >
> > ### **Q2: How is  VLB ensure or verified that the generated questions are not found in the pretraining data？**
> >
> > Tanks for your comment, we also believe this verification is important. We have conducted the experiment on image-text data contamination in section 5.4. We are pleased to provide you with more details and conducted additional experiments for you.
> >
> > **(1) As the section 5.4 and figure 8,** **we re-detect the contamination rate between the newly generated variants(v1+l4) and three pretraining datasets among MME, SEEDBench, and MMBench.** Using the same method we proposed in Section 3, we first identified the whether the new images are contaminated via clipscore, and then detect whether the corresponding new questions can be found or directly answered in pretraining captions. According to the results in figure 8, the image-text contamination rate of our dynamic variants has significantly decreases among all benchmarks and pretraining datasets, which proves that the proportion of new questions found in the pretraining data also significantly decreased.
> >
> > **(2) To further address your concerns, we detect the text contamination rates of the newly generated questions after strategy l1, l2,l3,l4**, based on the MME, SEEDBench, and MMBench benchmarks. Together with the origianl text contamination rate among contaminated images for comparison,  the experimental results are shown in the table below.
> >
> > | Pretraining-Dataset     | Benchmark | vanilla | L1     | L2     | L3     | L4     |
> > | ----------------------- | --------- | ------- | ------ | ------ | ------ | ------ |
> > | **LAION (100M)**        | SEEDBench | 0.6531  | 0.5197 | 0.4461 | 0.4286 | 0.4021 |
> > |                         | MMBench   | 0.8574  | 0.7846 | 0.7238 | 0.7165 | 0.6985 |
> > |                         | MME       | 0.9572  | 0.8540 | 0.8338 | 0.7434 | 0.5528 |
> > | **CC3M (3M)**           | SEEDBench | 0.3540  | 0.2622 | 0.2060 | 0.2165 | 0.1912 |
> > |                         | MMBench   | 0.6871  | 0.5327 | 0.4318 | 0.4018 | 0.4191 |
> > |                         | MME       | 0.6213  | 0.4984 | 0.5001 | 0.4314 | 0.3909 |
> > | **COCO-Caption (0.1M)** | SEEDBench | 0.9501  | 0.6952 | 0.6429 | 0.6354 | 0.5994 |
> > |                         | MMBench   | 0.8386  | 0.7410 | 0.6347 | 0.7143 | 0.7085 |
> > |                         | MME       | 0.8121  | 0.7227 | 0.6533 | 0.6892 | 0.7121 |
> >
> > Table B. The  text contamination rate about the original, and L1, L2, L3, L4 strategy between benchmarks and pre-training dataset.
> >
> > As can be seen,  the text contamination rates of our l1, l2,l3,l4 question variants have decreased on three pretraining sets LAION(100M) , CC3M(3M) and COCO-Caption (0.1M).  These proves that our newly generated questions are not found in the pretraining data.
> >
> >
> >
> >
> >
> >
> >
> > ### **References**
> >
> > [1] Wolfe, Jeremy M. "Visual attention." *Seeing* (2000): 335-386.
> >
> > [2] Xu, Jiang, et al. "Language in context: emergent features of word, sentence, and narrative comprehension." *Neuroimage* 25.3 (2005): 1002-1015.
> >
> > [3] Shah, Meet, et al. "Cycle-consistency for robust visual question answering." *Proceedings of the IEEE/CVF Conference on Computer Vision and Pattern Recognition*. 2019.
> >
> > [4] Gokhale, Tejas, et al. "Vqa-lol: Visual question answering under the lens of logic." *European conference on computer vision*. Cham: Springer International Publishing, 2020.
> >
> > [5] Selvaraju, Ramprasaath R., et al. "Squinting at vqa models: Introspecting vqa models with sub-questions." *Proceedings of the IEEE/CVF Conference on Computer Vision and Pattern Recognition*. 2020.
> >
> > [6] Chen, Xinlei, et al. "Microsoft coco captions: Data collection and evaluation server." *arXiv preprint arXiv:1504.00325* (2015).

---

> > > ### Comment · Reviewer_Rays · 2024-11-22
> > > **Reviewer Rays' response for Q2**
> > >
> > > Thank you. Table B is an important finding. Adding it to the paper (or supplementary) is recommended.

---

> > ### Comment · Reviewer_Rays · 2024-11-22
> > **Reviewer Rays' response for W2 and W3**
> >
> > W2: thank you. Drawing this distinction between prior related works will be useful.  I see that you have updated Sec 2 to reflect this.
> >
> > W3: thank you. I recommend adding these experiments (if possible at a larger scale) to the paper

---

> > > ### Author Response · Authors · 2024-11-26
> > > **Authors response for W3 updating**
> > >
> > > Dear Reviewer Rays,
> > >
> > > Thanks very much for your further expectation on W3!
> > >
> > > We take every comment you have made very seriously. According to your expectation, **these days we have enlarged the experiment on coco-caption as we responsed.** **We have increased the number of samples and also expanded the number of evaluated LVLMs.**
> > >
> > > Similarly, to verify VLB's capability in image caption task, we randomly sample a **500** pairs subset of coco-caption. Then, we use our three visual strategies to generate new images: adding new objects(v1), removing existing objects(v2), and outpainting(v3). Next, for each newly generated image, we use GPT-4V to generate corresponding captions, composing new <image, caption> pairs. Finally, we evaluate the original and the three dynamic COCO Caption datasets on **6** popular VLMs, through the image caption evaluating score integrated in Evalkit. The results are shown in the table below.
> > >
> > > | Model        | Vanilla | V1   | V2   | V3   |
> > > | ------------ | ------- | ---- | ---- | ---- |
> > > | TransCore-M  | 65.4    | 60.1 | 62.4 | 59.1 |
> > > | DeepSeek     | 58.6    | 46.1 | 51.0 | 45.2 |
> > > | InternVL-2   | 63.8    | 59.7 | 65.9 | 60.3 |
> > > | XComposer2   | 70.1    | 65.4 | 68.6 | 67.6 |
> > > | Monkey-Chat  | 68.5    | 61.6 | 65.9 | 62.8 |
> > > | Qwen-VL-Chat | 39.8    | 40.8 | 42.5 | 36.3 |
> > >
> > > Table C. The enlarged result of image captioning task about the original, and V1, V2, V3 strategy in COCO-Caption.
> > >
> > > As can be seen from the table, our newly generated samples achieve similar results to the original dataset among LVLMs. And similar conclusions can be drawn, namely that strategy V2 is simpler than V1 and V3. **This demonstrates that our dynamic framework can also effectively be applied to other multi-modal tasks, allowing a more flexible and comprehensive evaluation of LVLMs.** **We have also updated the content and results of the enlarged experiments in the revised paper, Appendix A.13 and Table 15.**
> > >
> > > Thanks for your interest in our research, which have been instrumental to our work. We are deeply grateful for your thorough and constructive review!

---

> > > > ### Comment · Reviewer_Rays · 2024-11-27
> > > > **increasing my rating**
> > > >
> > > > During the rebuttal phase my questions have been largely answered and I shall be increasing my rating shortly. Thanks to the authors for responding with additional results and insights that will strengthen the paper.

---

> > > > > ### Author Response · Authors · 2024-11-28
> > > > >
> > > > > Dear Reviewer Rays,
> > > > >
> > > > > Thanks very much for your thorough review and valuable comments, which have been instrumental in refining our paper. It is truly our luck to discuss with such a meticulous and knowledgeable reviewer as you!  We are deeply grateful!

---

> ### Author Response · Authors · 2024-11-23
> **Authors response for W1&Q1**
>
> Dear Reviewer Rays,
>
> Thanks very much for your response on W1&Q1!
>
> We are glad that our detailed explanations have addressed your concerns. We have now updated this content in Appendix A.11 of the revised paper constrained by page limitations. We believe our discussion with you has been invaluable, which will help readers better understand our motivation and contributions.
>
> Thanks for your thorough and constructive review! We are so lucky to have met such a careful and insightful reviewer like you!

---

> ### Author Response · Authors · 2024-11-23
> **Authors response for W2&W3**
>
> Dear Reviewer Rays,
>
> Thanks very much for your response on W2 and W3!
>
> We are glad that our additional experiments have addressed most of your concerns. And about W3, we will conduct a larger scale of experiments these days, and then update the experimental results for you. At that time, we will add this content into a new revised paper in a few days.
>
> We are deeply grateful for your thorough and constructive review!

---

> ### Author Response · Authors · 2024-11-23
> **Authors response for Q2**
>
> Dear Reviewer Rays,
>
> Thanks very much for your response on Q2!
>
> We are glad that our detailed experiments have addressed your concerns. We also regard table B important, and have been planning to update the content discussed with you into our paper. Due to the page limitations, we now updated this content in Appendix A.12 of the revised paper.
>
> Thanks for your interest in our research and for your constructive comments, which have been instrumental to our work!

---

### Author Response · Authors · 2024-11-21
**General Response：Thanks, Contributions, New Clarifications, and New Experiments**

We sincerely appreciate all reviewers’ time and efforts in reviewing our paper. We are glad that reviewers generally appreciate our proposed vision-language bootstrapping(VLB) for dynamic multi-modal evaluation. We begin by quantitatively detecting the degree of contamination present in current benchmarks, indicating the need for dynamic evaluation. Therefore with VLB, we can evolve existing benchmarks with visual and linguistic dynamics, obtaining various variants with flexible complexity and reduced data contamination.

### **Contributions**

Main contributions recognized by reviewers are concluded as follows:

- **Novelty.** The idea of using insights from user interactions to inform the transformations V and L is interesting. [Rays]; The paper proposes a novel dynamic multimodal evaluation framework VLB, which has a flexible complexity adjustment evaluation mechanism. [jqNx]
- **Analysis and Experiments.** The experiments are comprehensive.[Rays]; Extensive experiments are conducted to show performance change with VLB. [RXK7]; The VLB framework is highly versatile. It can be easily plugged in a lot of benchmarks.[jqNx]
- **Impact.** The paper identifies an important research direction.[Rays]; The new evaluation protocol can be readily combined with existing LVLM benchmarks. [RXK7]; VLB may help establish fairer baselines for LVLM evaluations.[7UPF]
- **Writing.** Good presentation and structured paper [7UPF, RXK7,  Rays].; Excellent, The paper is a pleasant read and is easy to follow, which is written in a well structured way following a clear plot line.[7UPF]

### **New Clarifications, and New Experiments**

We also thank all reviewers for their insightful and constructive suggestions, which helped a lot in further improving our paper. In addition to the pointwise responses below, we summarize supporting clarifications and experiments added in the rebuttal according to the reviewers’ suggestions.

**New Clarifications:**

- some detailed safeguard mechanisms to ensure the veracity and consistency of the VLB-generated samples. [7UPF, jqNx]
- some detailed visual attention and linguistic understanding interpretation of our dynamic strategy in user interaction. [Rays]
- The difference and overlap of our proposed linguistic rephrasing strategy with those other benchmarks. [Rays]
- The friendly cost needed in our framework VLB, about time, resources and evaluation. And the specific VLMs params. [jqNx]

**New Experiments and Analysis:**

- Applicability of our framework in image caption task. [Rays]
- Text-image contamination rates of the newly generated questions among pretraining data [Rays];
- The scale of the performance variance measured by standard deviation. [RXK7]
- Compatibility verification on image editing model. [RXK7]
- Ablation study and average adversarial iterations of the judge module. [7UPF]
- Some analysis of the evaluation results.[jqNx]
- Some analysis on the task categories of evaluation results.[jqNx]

We hope our pointwise responses below can clarify all reviewers’ confusion and alleviate all concerns. Thanks to all reviewers, and we have incorporated some mentioned contents into the revised paper, highlighted in blue.

---

### Meta-Review · Area_Chair_8W8n · 2024-12-19

**Metareview:**

This paper introduces a novel evaluation protocol, Vision-Language Bootstrapping (VLB), designed to comprehensively assess large vision-language models (LVLMs) while minimizing data contamination and enabling dynamic difficulty adjustment. Unlike existing benchmarks,  VLB allows both the image and the question to be modified. These modifications adjust the difficulty level and prevent models from relying on memorization, while maintaining consistency with the original answer. By applying various operations to alter the image or question, VLB generates a spectrum of difficulty levels. Extensive experimental results demonstrate the impact of these adjustments, presenting new challenges for the evaluation of LVLMs. The draft got positive reviews from all reviewers and the AC agrees and recommends an accept.

**Additional Comments On Reviewer Discussion:**

Great discussion with detailed responses from the authors. Reviewers are positive with the new results.

---

### Decision · Program_Chairs · 2025-01-22

Accept (Oral)